# Unbiased screening identifies regulators of cell-cell adhesion and treatment options in pemphigus

Henriette Franz [1], Maitreyi Rathod [1,2], Aude Zimmermann [1], Chiara Stüdle[1,3], Vivien Beyersdorfer[1,2], Karen Leal-Fischer [1], Pauline Hanns[1], Tomás Cunha[4], Dario Didona[4], Michael Hertl[4], Marion Scheibe[5,6], Falk Butter [5,6], Enno Schmidt [7] & Volker Spindler [1,2] ✉

Cell-cell junctions, and specifically desmosomes, are crucial for robust intercellular adhesion. Desmosomal function is compromised in the autoimmune blistering skin disease pemphigus vulgaris. We combine whole-genome knockout screening and a promotor screen of the desmosomal gene desmoglein 3 in human keratinocytes to identify novel regulators of intercellular adhesion. Kruppel-like-factor 5 (KLF5) directly binds to the desmoglein 3 regulatory region and promotes adhesion. Reduced levels of KLF5 in patient tissue indicate a role in pemphigus vulgaris. Autoantibody fractions from patients impair intercellular adhesion and reduce KLF5 levels in in vitro and in vivo disease models. These effects were dependent on increased activity of histone deacetylase 3, leading to transcriptional repression of KLF5. Inhibiting histone deacetylase 3 increases KLF5 levels and protects against the deleterious effects of autoantibodies in murine and human pemphigus vulgaris models. Together, KLF5 and histone deacetylase 3 are regulators of desmoglein 3 gene expression and intercellular adhesion and represent potential therapeutic targets in pemphigus vulgaris.

Dynamic contact of neighboring cells is essential to form and maintain multicellular organisms and to provide barriers and resilience against external cues[1]. In vertebrates, a specific type of cell–cell contact, the desmosome, evolved[2]. Desmosomes facilitate strong intercellular adhesion due to their molecular composition and linkage to the intermediate filament cytoskeleton[3,4]. Accordingly, desmosomes are most abundant in tissues exposed to a high degree of mechanical stress. In particular, the epidermis, the epithelia of the oral and vaginal cavity, and the myocardium rely on large numbers of desmosomes to withstand the multitude of forces acting on the tissue[4,5]. The general blueprint of desmosomes is similar throughout the body: The core is built up of transmembrane cadherin-type adhesion molecules, the desmogleins (DSG) and desmocollins (DSC). These are coupled to the intermediate filament cytoskeleton by a set of adapter proteins consisting of plakoglobin, plakophilins (PKP), and desmoplakin (DSP). DSGs, DSCs, and PKPs comprise different isoforms, which show tissue-specific distribution. For example, stratifying epithelia such as the epidermis show a predominance of DSG1 and DSG3, while in simple epithelia and the myocardium the expression is restricted to DSG2.

A plethora of diseases and developmental defects are associated with dysfunctional adhesion[6,7]. With regard to desmosomes, autoantibodies targeting DSG1 and DSG3, which impair intercellular

[1]Department of Biomedicine, University of Basel, Basel, Switzerland. [2]Institute of Anatomy and Experimental Morphology, University Medical Center Hamburg Eppendorf (UKE), Hamburg, Germany. [3]Theodor Kocher Institute, University of Bern, Bern, Switzerland. [4]Klinik für Dermatologie und Allergologie, Philipps-Universität Marburg, Marburg, Deutschland. [5]Institute of Molecular Biology (IMB), Mainz, Germany. [6]Institute of Molecular Virology and Cell Biology, Friedrich-Loeffler-Institute, Greifswald, Germany. [7]Department of Dermatology, University of Lübeck, Lübeck, Germany; Lübeck Institute for Experimental Dermatology, University of Lübeck, Lübeck, Germany. ✉e-mail: v.spindler@uke.de

adhesion, cause the autoimmune skin blistering diseases pemphigus vulgaris (PV) and pemphigus foliaceus[8]. Mutations in any of the desmosomal proteins, in particular PKP2, DSG2, and desmoplakin, cause Arrhythmogenic cardiomyopathy, a disease characterized by arrhythmia up to sudden cardiac death and functional impairment of the heart[9]. It was recently shown that indeed loss of adhesion appears to be an early step for disease development[10].

Given these causal relationships between altered desmosome function, impaired adhesion and disease, modulation of desmosomal function may represent a relevant strategy for targeted treatment. However, the mechanisms regulating expression, trafficking, and membrane positioning of the molecular components of desmosomes, ultimately leading to cellular adhesive properties fit for the actual demand, are only partially understood.

In the current study, we used unbiased screening approaches to elucidate hitherto unknown pathways that regulate desmosomal adhesion. We then focused on the role of HDAC3 via KLF5 to modulate DSG3 levels and adhesion, a regulatory axis we show to be impaired in PV. Inhibiting HDAC3 prevented the loss of intercellular adhesion in vitro and blister formation in vivo in the context of PV.

## Results

### Identification of putative cell–cell adhesion regulators
To uncover modulators of intercellular adhesion, we performed a whole-genome CRISPR knockout screen (Fig. 1a) using the human HaCaT keratinocyte cell line. To do so, we first established a single-cell clone stably expressing Cas9 at similar functional levels. The Cas9 editing efficiency was verified by transduction with a lentivirus containing the reporter pKLV2-U6gRNA5(gGFP)-PGKBFP2AGFP-W[11] followed by flow cytometry. The selected cell line showed an editing efficiency of 99.46% (Supplementary Fig. 1a). Cells were transduced with a sgRNA library covering 19.114 genes of the human genome, with 4 individual sgRNAs per gene[12]. Because interference with the amount and stability of desmosomal adhesion molecules at the cell membrane of epidermal keratinocytes or mucosal epithelial cells leads to loss of intercellular adhesion in pemphigus[13] we chose to evaluate membrane levels of the adhesion molecule DSG3 and use these as proxy for intercellular adhesion. The top and bottom ten percent of cells with the highest and lowest DSG3 levels were sorted, and the sgRNA loci were amplified and sequenced individually for each pool. SgRNAs that target genes relevant for high DSG3 membrane levels should be enriched in the DSG3^{low} pool while those sgRNAs that target molecules impairing DSG3 membrane levels are expected to be abundant in the DSG3^{high} pool. Potential positive regulators of DSG3 were identified by comparing the DSG3^{low} with the DSG3^{high} pool (Fig. 1b, Supplementary Table 1). A gene ontology pathway analysis for biological processes revealed significant enrichment in diverse groups such as "gene expression", "cellular component organization" and "cellular membrane organization" (Supplementary Fig. 1b). The identification of *DSG3* among the highest-ranking hits supported the robustness of our

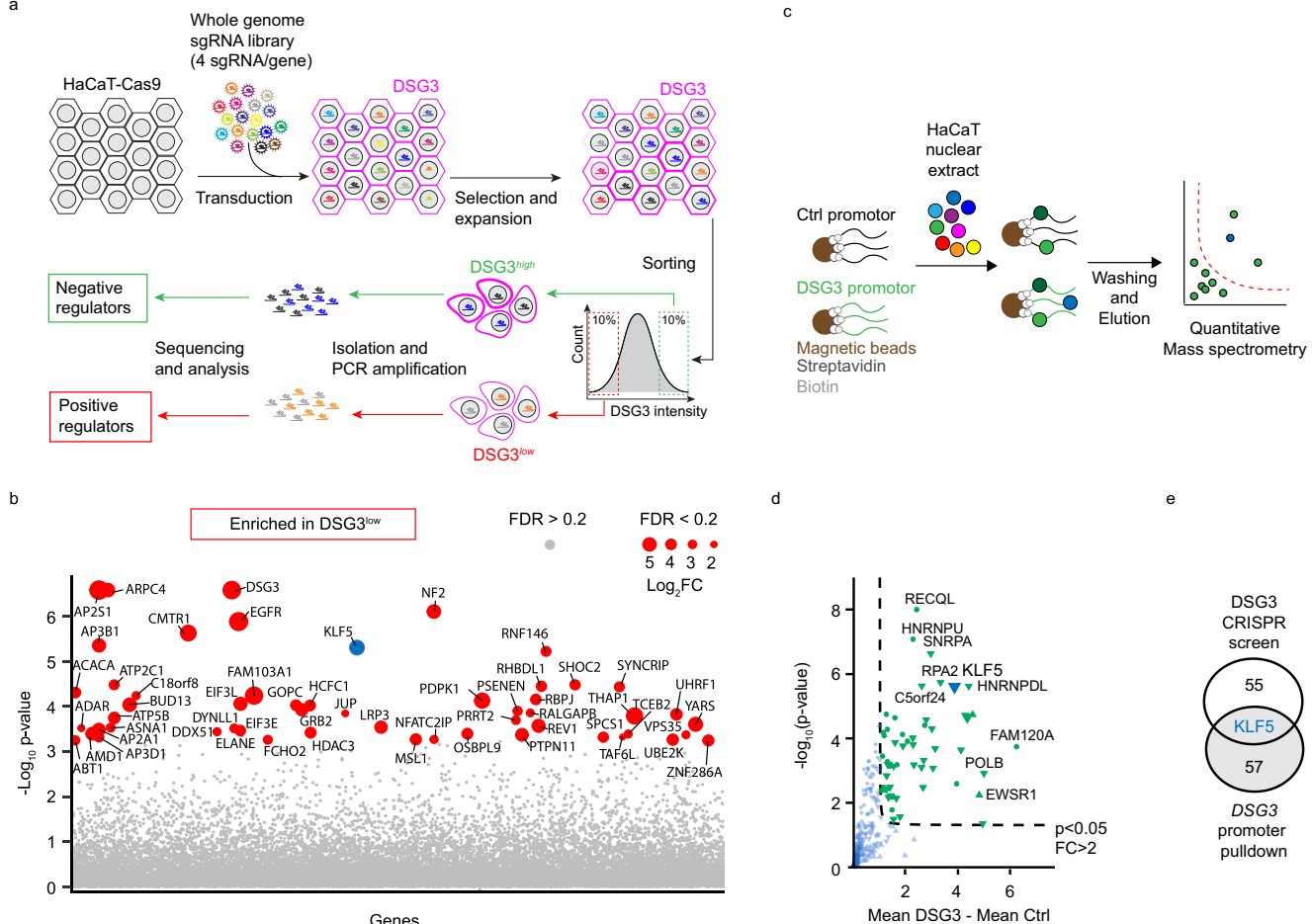

**Fig. 1 | Identification of KLF5 as a positive regulator for DSG3. a** Scheme of the CRISPR/Cas9 sgRNA library screen. **b** Scatterblot showing genes and according to p-values by comparison of the DSG3^{low} versus DSG3^{high} pools. Fold changes are indicated by the dot size. **c** Scheme of DSG3 promoter screen. **d** Scatterblot of genes enriched at the DSG3 promoter blotted according to fold change and p-value. **e** Overlap of positive regulators identified in CRISPR/Cas9 sgRNA library screen and proteins binding to the DSG3 promoter identified by promoter pulldown. Source data are provided as a Source Data file.

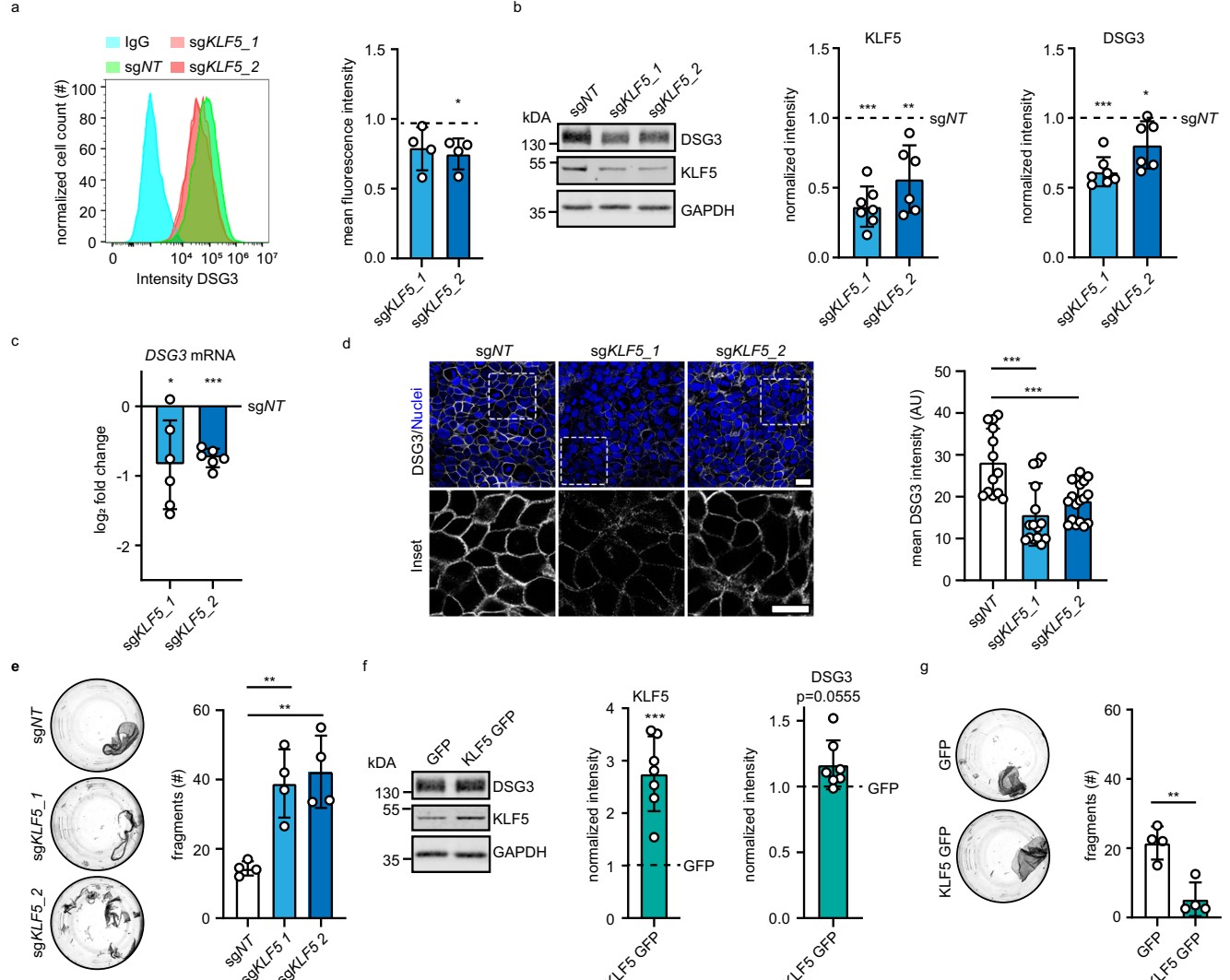

**Fig. 2 | KLF5 modulates DSG3 protein levels and cell−cell adhesion. a** Flow cytometry analysis of HaCaT cells stained with anti-DSG3 antibodies or IgG as control. HaCaT cells were transduced with sg*NT*, sg*KLF5_1* (p = 0.0698), or sg*KLF5_2* (p = 0.0206). A representative histogram and mean fluorescence intensity of four independent experiments are displayed. **b** Western blot analysis of HaCaT cell lysates using KLF5, DSG3, and GAPDH antibodies. HaCaT cells were stably transduced with sg*NT*, sg*KLF5_1*, or sg*KLF5_2*. Representative Western blot images and quantifications of the respective proteins are shown; sg*KLF_1* (p = 0.0005, n = 7), sg*KLF5_2* (p = 0.0023, n = 6). Values were normalized to sg*NT*. **c** Quantitative Real-time PCR analysis of *DSG3* of mRNA extracted from HaCaT cells stably transduced with sg*NT*, sg*KLF5_1*, or sg*KLF5_2*. Values were normalized to sg*NT*. n = 6, sg*KLF5_1* p = 0.0235; sg*KLF5_2* p < 0.0001. **d** Immunofluorescence staining of HaCaT cells stably transduced with sg*NT*, sg*KLF5_1* or sg*KLF5_2* using DSG3 antibodies and DAPI as nuclear stain (n = 3, sg*KLF5_1* p < 0.0001, sg*KLF5_2* p = 0.0009). Representative pictures are shown. Scale bar = 10 μm. **e** Dispase-based dissociation assay of HaCaT cells stably transduced with sg*NT*, sg*KLF5_1*, or sg*KLF5_2* (n = 4, sg*KLF5_1* p = 0.0046; sg*KLF5_2* p = 0.0020). Representative images and the number of fragments are shown. **f** Western blot analysis of HaCaT cell lysates stably overexpressing KLF5 using KLF5, DSG3, and GAPDH antibodies. Representative Western blot images and quantifications of respective proteins (n = 7, KLF5 p = 0.0006, DSG3 p = 0.0555), are shown. **g** Dispase-based dissociation assay of HaCaT cells stably overexpressing KLF5 (n = 4, p = 0.0031). Representative images and the number of fragments are shown. Values expressed as mean with standard deviation (mean ± SD). One n represents one biological replicate. Source data are provided as a Source Data file. Experiments (**a**–**c**, **f**) were analyzed with one-sample t-test (two-sided), (**d**, **e**) were analyzed with One-way-ANOVA, SIDAK correction. **g** was analyzed with student's t-test (two-sided). p < 0.05*; p < 0.01**; p < 0.001***.

screen (Fig. 1b, Supplementary Table 1). Transduction of a parental, non-clonal HaCaT line with a sgRNA targeting DSG3 in a different vector (LentiCRISPR v2 containing Cas9) similarly led to significantly less DSG3 at the membrane and resulted in reduced intercellular adhesion as tested by dispase-based dissociation assays (Supplementary Fig. 1c, d), confirming that loss of membrane DSG3 causes impaired adhesion. Next, eight other high-ranking positive regulators were selected for verification and further analysis, and stable knock-outs for each gene were generated in HaCaTs using individual sgRNAs cloned into LentiCRISPR v2. While our results with sgRNAs against DSG3 demonstrated that indeed membrane levels of this protein and

cell−cell adhesion correlate in HaCaT, this was only partially the case for the selected targets (Supplementary Fig. 1e−g). To combine these results with direct regulators of DSG3, we additionally performed a screen for DSG3 transcriptional regulators. The DSG3 promotor was cloned and amplified with biotinylated primers, attached to streptavidin-coated beads, and incubated with HaCaT nuclear extracts. The promotor-bound protein fraction was then analyzed by quantitative mass spectrometry (Fig. 1c, d, Supplementary Table 2). Interestingly, overlapping the results of both screens revealed the transcription factor Kruppel-like-factor 5 (KLF5) as both a positive regulator of DSG3 and a binder to the DSG3 promoter region (Fig. 1e).

KLF5 binding appeared to be specific to the DSG3 promoter as the proteins that co-precipitated with the DSG2 promoter, despite a profound overlap of hits, did not include KLF5 (Supplementary Fig. 2a, b; Supplementary Table 3). To verify that KLF5 binds to the DSG3 promoter, we performed KLF5 ChIP-qPCR and showed that KLF5 recognizes a DNA region close to the transcription start site of DSG3 (Supplementary Fig. 2c). CTNNB1, a known KLF5 target[14], served as positive control. Analysis of a published HaCaT ChIP sequencing dataset (GEO: GSE168600)[15] supported these results (Supplementary Fig. 2d).

## KLF5 is a positive regulator of DSG3 and intercellular adhesion

Flow cytometry of two HaCaT lines transduced with two individual sgRNAs targeting KLF5 confirmed that KLF5 depletion resulted in reduced DSG3 membrane levels, reduced total DSG3 levels, and reduced mRNA transcript abundance (Fig. 2a–c), suggesting that KLF5 activates DSG3 transcription. A decrease of DSG3 protein levels was also observed in KLF5-depleted normal human epidermal keratinocytes (NHEK, Supplementary Fig. 3a). Furthermore, immunofluorescence staining of DSG3 in HaCaT cells showed that DSG3 membrane localization is reduced in KLF5-depleted cells (Fig. 2d). Dissociation assays revealed that depletion of KLF5 caused reduced cell–cell adhesion in HaCaT cells (Fig. 2e) and NHEKs (Supplementary Fig. 3b). Conversely, overexpression by transduction of KLF5-GFP led to an increase of DSG3 protein levels and enhanced cell–cell adhesion in HaCaT cells (Fig. 2f, g). These results demonstrate a regulation of DSG3 transcription, DSG3 membrane levels, and intercellular adhesion by KLF5.

## KLF5 and HDAC3 are altered in pemphigus vulgaris

We next tested whether KLF5 is relevant for intercellular adhesion in disease settings. To uncover potential alterations in PV, we first performed immunostaining of human skin and mucosa sections of PV patients versus healthy controls. Interestingly, KLF5 levels were reduced in both PV patient sample groups (Fig. 3a, b, Supplementary Table 4), suggesting a contribution to disease. To further explore this possibility, we incubated HaCaT cells with PV-IgG, which led to binding of the antibodies to the cell surface (Supplementary Fig. 4a, b) and a significant reduction of DSG3 protein levels (Fig. 3c). Importantly, KLF5 levels significantly decreased upon PV-IgG treatment (Fig. 3c). These findings were confirmed in NHEKs using PV-IgG and px4_3, a single chain variable fragment (scFv) cloned from a PV patient B-cell repertoire and targeting both DSG3 and DSG1[16] (Supplementary Fig. 4c). Expression of KLF5-GFP rescued the levels of DSG3 protein and reduced the loss of cell–cell adhesion in response to PV-IgG (Fig. 3d, Supplementary Fig. 4d, e), suggesting a contribution of KLF5 to disease.

The activity of KLF transcription factors can be regulated by acetylation and through histone deacetylases (HDACs)[17–19]. Indeed, we detected acetylation of KLF5 in HaCaT cells using Co-IP experiments (Supplementary Fig. 4f). Moreover, PV-IgG treatment of HaCaT cells resulted in a significant increase in HDAC activity (Fig. 3e) and HDAC3 protein levels (Fig. 3f). HDAC activity was fully blocked by trichostatin A, an inhibitor of class I and II HDACs. In contrast, RGFP966, a selective HDAC3 inhibitor, only blocked the activity increase by PV-IgG, suggesting that HDAC3 is the primary HDAC activated in response to autoantibodies. In line with these results, immunofluorescence analysis showed a significant increase of HDAC3 in human skin and mucosa of pemphigus vulgaris patients (Fig. 3g, h).

To examine whether high HDAC3 expression and activity is sufficient to cause a general loss of cell–cell adhesion, we overexpressed HDAC3 in HaCaT cells (Supplementary Fig. 5a). HDAC3-GFP transduction resulted in the expected increase in HDAC3 activity (Supplementary Fig. 5b) and a decrease in cell–cell adhesion (Supplementary Fig. 5c), without affecting DSG3 levels. However, the transduction of

HaCaT cells with sgRNA targeting HDAC3 resulted in a significant reduction of DSG3 protein levels and cell–cell adhesion (Supplementary Fig. 5d, e). This is in line with the results of the CRISPR screen, which suggested HDAC3 as a positive regulator of DSG3 in the membrane (Fig. 1b). As both overexpression as well as depletion of HDAC3 resulted in reduced intercellular adhesion, a precisely balanced activity of HDAC3 appears to be required for physiological levels of cell cohesion.

## P38MAPK-dependent HDAC3 increase negatively regulates KLF5 transcription in PV-IgG-treated keratinocytes

Although PV-IgG led to higher levels of HDAC3 and reduced levels of KLF5, in line with a pathway of impaired KLF5 acetylation by HDAC3, such a mechanism is not supported by our data. PV-IgG treatment did not significantly alter KLF5 acetylation as revealed by an acetylation-specific KLF5 antibody (Fig. 4a). Nevertheless, KLF5 acetylation may still play a role in the regulation of intercellular adhesion, as expression of wildtype but not the acetylation-deficient mutants KLF5 K369A and KLF5 K391A was effective to prevent loss of cell–cell adhesion in response to PV-IgG (Supplementary Fig. 6a).

Alternatively, HDAC3 may act through its role of deacetylating histones, leading to compacted chromatin and reduced transcription of genes in these regions[20]. Interestingly, PV-IgG treatment led to a reduction of KLF5 mRNA (Fig. 4b) as detected with four independent primer pairs, suggesting a regulation of KLF5 at the transcriptional level, i.e., as a negative regulator of the *KLF5* gene. In line with this hypothesis, depletion of HDAC3 in HaCaT cells led to a significant increase of *KLF5* mRNA (Fig. 4c). PV-IgG treatment in HDAC3-depleted cells did not change KLF5 protein levels (Fig. 4d, e). Moreover, ChIP followed by quantitative real-time PCR showed HDAC3 to be significantly enriched at the *KLF5*, but not the *DSG3* promoter (Fig. 4f). These data were confirmed by analysis of published HDAC3 ChIP sequencing results (GEO: GSE137232, Supplementary Fig. 6b)[21]. Furthermore, treatment with PV-IgG led to an increase of HDAC3 at the KLF5 promoter (Fig. 4g). This suggests that increased levels of HDAC3 in response to PV-IgG repress KLF5 and subsequently DSG3 transcription. Vice versa, depletion of DSG3 in HaCaT cells did not result in changes of either HDAC3 or KLF5, showing that these proteins act upstream of DSG3 (Supplementary Fig. 6c). To better understand how PV-IgG induce HDAC3 levels, we tested the interference with p38MAPK activity, as a rapid activation of this kinase by PV-IgG is well established[22–24]. Interestingly, inhibition of p38MAPK with SB202190 reduced HDAC3 levels in PV-IgG-treated HaCaT cells (Fig. 4h), suggesting that activated p38MAPK promotes HDAC3 protein levels in PV.

Taken together, PV-IgG led to increased HDAC3 protein levels followed by a depletion of KLF5 and DSG3. These effects appeared to be specific for DSG3, as inhibition of HDAC3 with RGFP966[25] did not lead to changes in other desmosomal cadherins or DSP (Supplementary Fig. 6d).

## HDAC3 inhibition ameliorates PV-IgG-induced loss of cell–cell adhesion in vitro

The increased activity of HDAC3 and decreased levels of KLF5 in response to PV-IgG incubation suggests blocking of HDAC3 as potential treatment strategy. To investigate if HDAC3 inhibition prevents PV-IgG-induced loss of cell–cell adhesion, we incubated HaCaT cells with PV-IgG and titrated the HDAC3-specific inhibitor RGFP966. The application of 1 or 5 μM RGFP966 for 24 h restored cell–cell-adhesion in PV-IgG-treated cells (Fig. 5a). We confirmed these results using the HDAC1/HDAC3 inhibitor Entinostat (Supplementary Fig. 7a). Similarly, loss of cell–cell adhesion caused by pX4_3 was ameliorated by different concentrations of RGFP966 (Supplementary Fig. 7b, c). In addition, incubation of PV-IgG-treated NHEK cells with RGFP966 ameliorated loss of intercellular adhesion (Supplementary Fig. 7d). These effects

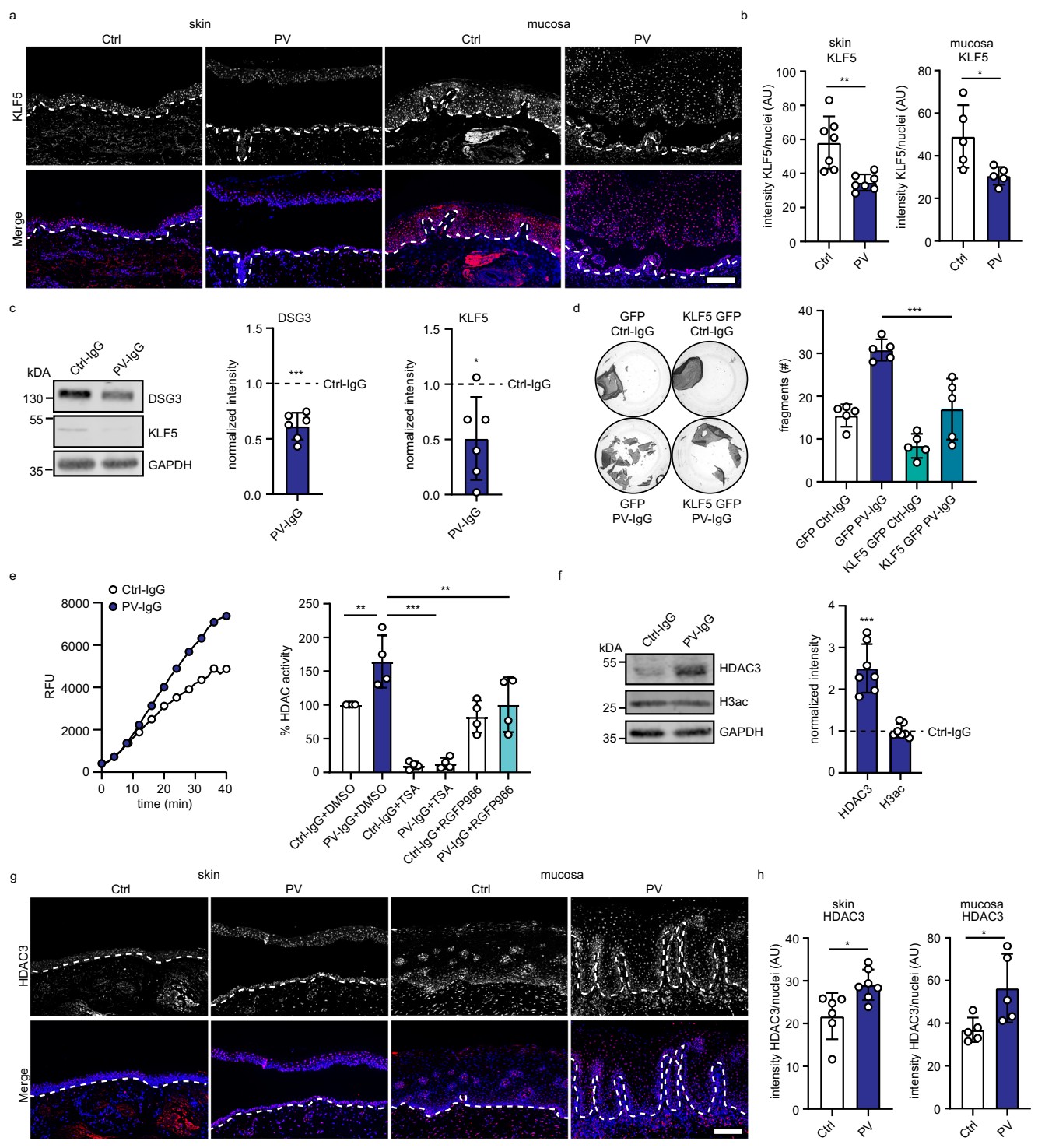

were dependent on KLF5, as HDAC3 inhibition failed to restore inter-cellular adhesion in KLF5-depleted HaCaT cells (Supplementary Fig 7e). Furthermore, sgRNA-mediated silencing of HDAC3 reduced susceptibility to PV-IgG-mediated loss of cell–cell adhesion (Supplementary Fig. 7f). DSG3 protein levels were restored upon RGFP treatment (Fig. 5b) and inhibition of HDAC3 with RGFP966 or Entinostat activated DSG3 promoter activity (Fig. 5c). Again, this was prevented by sgRNA-mediated KLF5 silencing (Supplementary Fig. 7g). Immu-nostaining for DSG3 and DSP showed that both RGFP966 and Entinostat treatment of HaCaT cells led to a restoration of DSG3 levels in the membrane (Fig. 5d) while DSP localization and levels remained unchanged. Similar results were obtained in NHEK cells

(Supplementary Fig. 8). Together, these results demonstrate that HDAC3 inhibition effectively reduced the PV-IgG phenotype in vitro.

## HDAC3 inhibition alleviates the PV phenotype in vivo

To uncover if HDAC3 inhibition is a viable treatment option also in vivo, we first applied the passive transfer neonatal mouse model[26]. We subcutaneously injected newborn 1-day-old neonates into the back skin with 40 μg of purified pX4_3 or control IgG. pX4_3 bound to the cell surface (Supplementary Fig. 9a) after 24 h. Neonates showed intraepidermal split formation in pX4_3-treated animals, which was absent in controls (Fig. 6a). In the neonate mouse model, co-injection of 5 μM RGFP966 with pX4_3 resulted in a significant reduction of

**Fig. 3 | KLF5 is downregulated in pemphigus vulgaris (PV).**
**a** Immunofluorescence staining of human skin or mucosa sections from healthy controls or PV biopsies using KLF5 antibodies and DAPI. Representative images of one individual patient and control are shown. Scale bar = 100 μm. **b** Quantification of (**a**) displaying the mean intensity of KLF5 per nucleus (skin n = 7 p = 0.0023, mucosa n = 5 p = 0.0270). **c** Western blot analysis of HaCaT cells treated with Ctrl-IgG or PV-IgG for 24 h using KLF5, DSG3, and GAPDH antibody. Representative Western blot images and quantifications of respective proteins (n = 6, DSG3 p = 0.0006, KLF5 p = 0.0238) are shown. Values were normalized to Ctrl-IgG. **d** Dispase-based dissociation assay of HaCaT cells stably transduced with GFP or KLF5-GFP (n = 5, GFP PV-IgG vs KLF5-GFP PV-IgG p < 0.0001). Representative images and the number of fragments are shown. **e** HDAC activity assay using HaCaT cell lysates. HaCaT cells were treated with Ctrl-IgG or PV-IgG including DMSO, 1 μM trichostatin A (TSA), or 5 μM RGFP966 for 24 h. The left panel shows the increase in deacetylated peptide detected by its fluorophore (expressed by RFU relative fluorescence units) over time of a representative experiment and the right panel

shows the calculated HDAC3 activity (n = 4, Ctrl-IgG+DMSO vs PV-IgG DMSO p = 0.0061, PV-IgG DMSO vs PV-IgG TSA p < 0.0001, PV-IgG DMSO vs PV-IgG RGFP966 p = 0.0063). **f** Western blot analysis of HaCaT cell lysates using HDAC3, H3ac, and GAPDH antibodies. HaCaT cells were treated with Ctrl-IgG or PV-IgG for 24 h. Representative Western blot images and quantifications of respective proteins (n = 7, HDAC3 p = 0.0005, H3ac p = 0.7207) are shown. Values were normalized to Ctrl-IgG. **g** Immunofluorescence staining of human skin or mucosa sections from healthy controls or PV biopsies using HDAC3 antibodies and DAPI. Representative images of individual patients and controls are shown. Scale bar = 100 μm. **h** Quantification of (**g**) displaying the mean intensity of HDAC3 per nucleus (skin n = 7 p = 0.0139, mucosa n = 5 p = 0.0335). Values expressed as mean with standard deviation (mean ± SD). One n represents one biological replicate. Source data are provided as a Source Data file. Experiments (**d**) and (**e**) were analyzed with One-way-ANOVA, SIDAK correction. **b**, **h** were analyzed with two-sided students t-test, and **c**, **f** were analyzed with one-sample t-test. p < 0.05*; p < 0.01**; p < 0.001***.

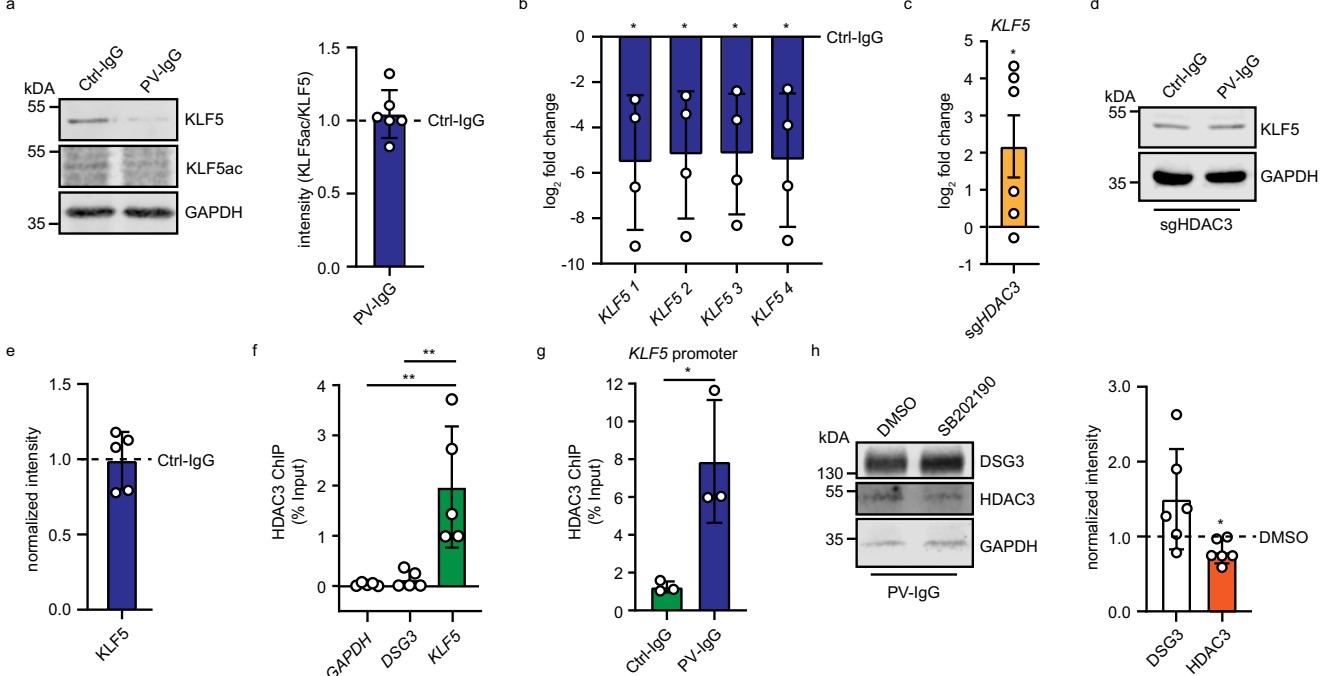

**Fig. 4 | P38MAPK-dependent HDAC3 increase negatively regulates KLF5 expression in PV-IgG-treated keratinocytes. a** Western blot analysis of HaCaT cells which were treated with Ctrl-IgG or PV-IgG for 24 h. Antibodies against KLF5, KLF5ac, and GAPDH were used. Representative Western blot images and quantifications of respective proteins (n = 6) are shown. Values were normalized to Ctrl-IgG. **b** Quantitative real-time PCR of mRNA extracted from HaCaT cells treated with IgG or PV-IgG. Four different primer pairs detecting KLF5 mRNA were used. Values were normalized to IgG control (n = 4, *KLF5*_1 p = 0.0334, *KLF5*_2 p = 0.0340, *KLF5*_3 p = 0.0301, *KLF5*_4 p = 0.0344). **c** Quantitative real-time PCR analysis of mRNA extracted from HaCaT cells stably transduced with sg*NT* or sg*HDAC3*. Primers detecting *KLF5* mRNA were used. Values were normalized to sg*NT* (n = 6, *KLF5* p = 0.0489) **d, e** Western Blot analysis of HaCaT cells stably transfected with sgHDAC3 treated with IgG or PV-IgG using DSG3, KLF5 and GAPDH antibodies.

Values were normalized to Ctrl-IgG. Representative images are shown (n = 5). **f** ChIP-quantitative real-time PCR analysis of HaCaT cells using HDAC3 antibodies and primers detecting *GAPDH*, *DSG3*, and *KLF5* promoters. (n = 5, *GAPDH vs KLF5* p = 0.0019, *DSG3 vs KLF5* p = 0.0028). **g** ChIP-quantitative real-time PCR analysis of HaCaT cells treated for 24 h with IgG or PV-IgG using HDAC3 antibodies and primers to detect the *KLF5* promoter (n = 3, Ctrl-IgG vs PV-IgG, p = 0.0245). **h** Western blot analysis of HaCaT cell lysates treated with PV-IgG and DMSO or SB202190 using DSG3, KLF5, HDAC3, and GAPDH antibodies. Values were normalized to DSMO (n = 6, DSG3 p = 0.1276, KLF5 p = 0.039). Representative images are shown. Values expressed as mean with standard deviation (mean ± SD). One n represents one biological replicate. Source data are provided as a Source Data file. **a**–**e**, **h** were analyzed with two-sided one-sample t-tests. **f** was analyzed with with One-way-ANOVA, SIDAK correction. **g** was analyzed with two-sided Students t-test. p < 0.05*.

pX4_3-induced blisters as did Entinostat at a concentration of 10 μM (Fig. 6b). To determine whether HDAC3 inhibition is also effective in human skin, we utilized an ex vivo human skin model[27]. Human skin from explants was subcutaneously injected with 40 μg of purified pX4_3 or IgG as a control. PX4_3 was detectable bound to the cell surface of keratinocytes as identified with an anti-HA antibody (Supplementary Fig. 9b). pX4-3 induced intraepidermal blistering after 24 h, which was significantly reduced when RGFP966 or Entinostat were co-injected (Fig. 6c, d). By immunostaining, we detected that

HDAC3 levels were increased in response to pX4_3 similar to the patient situation but were not altered by RGFP966 or Entinostat (Supplementary Fig. 9c, d). However, the reduced levels of KLF5 in pX4_3-injected skin were restored upon HDAC3 inhibition (Fig. 6e, f). Importantly, injection of RGFP966 or Entinostat 2 h after blister induction with pX4_3 also ameliorated the PV phenotype (Supplementary Fig. 9e, f). DSG3 protein levels were significantly reduced by pX4_3 and restored by RGFP966 or Entinostat treatment in human skin (Supplementary Fig. 10a, b). Finally, we applied an ex vivo PV model of

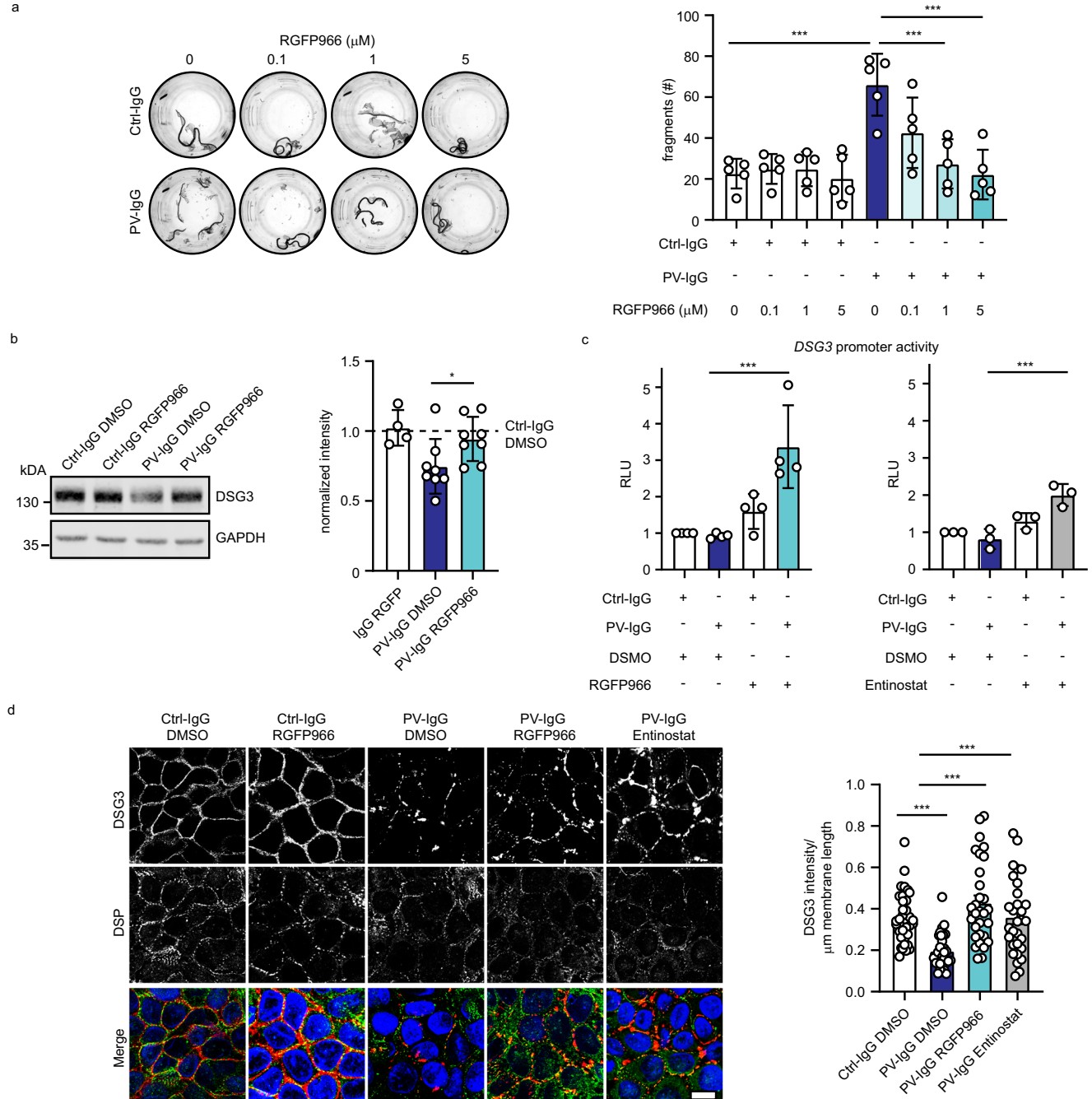

**Fig. 5 | HDAC3 inhibition prevents the PV-IgG-induced phenotype in vitro.**
**a** Dispase-based dissociation assay of HaCaT cells treated with Ctrl or PV-IgG and indicated concentrations of HDAC3 inhibitor RGFP966. Representative images and quantifications of (n = 5, Ctrl-IgG vs PV-IgG p < 0.0001, PV-IgG vs PV-IgG 1 μM RGFP966 p < 0.0001, PV-IgG vs PV-IgG 5 μM RGFP966 p < 0.0001) are shown.
**b** Western blot analysis of HaCaT cell lysates using DSG3 and GAPDH antibodies. HaCaT cells were treated for 24 h with Ctrl-IgG or PV-IgG and DMSO or RGFP966. Representative Western blot images and quantifications of respective proteins (Ctrl-IgG DMSO n = 8, Ctrl-IgG RGFP966 n = 4 , PV-IgG DMSO n = 8, PV-IgG RGFP966 n = 8, PV-IgG DMSO vs PV-IgG 1 μM RGFP966 p = 0.0340) are shown. Values were normalized to IgG DMSO. **c** Luciferase assay (luciferase activity expressed in RLU - relative luminescence units) of HaCaT cells treated with Ctrl-IgG or PV-IgG and

HDAC3 inhibitors 20 μM RGFP966 (n = 4, PV-IgG DMSO vs PV-IgG RGFP966 p = 0.0002) or 20 μM Entinostat (n = 3, PV-IgG DMSO vs PV-IgG Entinostat p = 0.0004). **d** Immunofluorescence staining of HaCaT cells treated with Ctrl-IgG or PV-IgG and 5 μM RGFP966 or 10 μM Entinostat using DSG3, DSP antibodies and DAPI. Scale bar = 10 μm. Quantification of DSG3 intensity/membrane length of 3 independent experiments are shown. Each data point represents one cell. PV-IgG DMSO vs Ctrl-IgG DMSO p = 0.0001, PV-IgG DMSO vs PV-IgG RGFP966 p < 0.0001, PV-IgG DMSO vs PV-IgG Entinostat p < 0.0002. Values are expressed as mean with standard deviation (mean + /-SD). One n represents one biological replicate. Source data are provided as a Source Data file. All experiments were statistically analyzed with One-way-ANOVA, SIDAK correction. p < 0.05*; p < 0.01**; p < 0.001***.

oral human mucosa to test the applicability of this approach to tissues affected in PV different from the epidermis[28]. PX4_3 resulted in pronounced intraepithelial blister formation after 24 h. Importantly, mucosal blistering was ameliorated by co-injection of RGFP966 or

Entinostat (Supplementary Fig. 10c, d). Together, HDAC3 inhibition prevented blistering in vivo and ex vivo in three independent PV models. This suggests HDAC3 as a promising option for targeted treatment of PV patients.

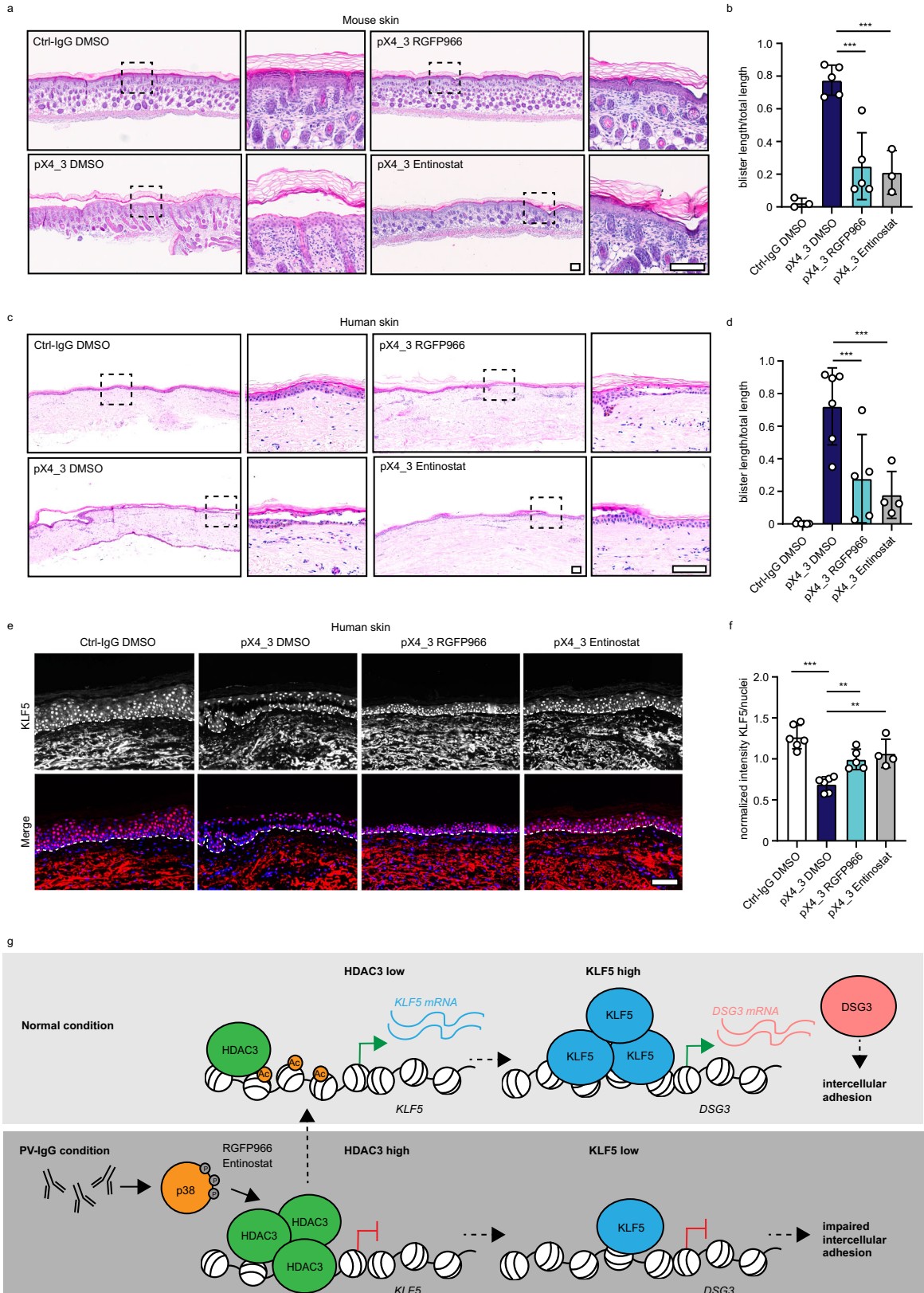

## Discussion

In our study, we combined a CRISPR/Cas9 knockout screen and a DSG3 promoter screen to detect regulators of desmosomal adhesion. Using this approach, we identified a disease-relevant pathway of DSG3 transcriptional regulation through HDAC3 and KLF5 (Fig. 6g, Supplementary Table 6). Under normal conditions, KLF5 is required for DSG3 transcription and strong intercellular adhesion. In PV conditions, HDAC3 activity is enhanced by PV-IgG via p38MAPK, leading to increased binding to the KLF5 promotor and repressing KLF5 transcription. Blocking of HDAC3 activity enhances Dsg3 expression by increasing KLF5 levels, leading to elevated DSG3 levels at the membrane with subsequent strengthening of intercellular adhesion.

**Fig. 6 | HDAC3 inhibition ameliorates PV phenotype in vivo. a** Passive transfer neonatal mouse model: H&E staining of mouse skin injected with Ctrl-IgG or pX4_3 and DMSO/ 5 μM RGFP966 or 10 μM Entinostat is displayed. Scale bar = 100 μm. **b** Quantification of blister length versus total section length is shown (Ctrl-IgG DMSO n = 3, pX4_3 DMSO n = 5, pX4_3 RGFP966 n = 5, pX4_3 Entinostat n = 3, pX4_3 DMSO vs pX4_3 RGFP966 p = 0.0001, pX4_3 DMSO vs pX4_3 Entinostat p = 0.0003). **c** Ex vivo human skin model: H&E staining of skin explants injected with Ctrl-IgG or pX4_3 and DMSO/ 5 μM RGFP966 or 10 μM Entinostat is displayed. Scale bar = 100 μm. **d** Quantification of blister length versus total section length is shown (Ctrl-IgG DMSO n = 6, pX4_3 DMSO n = 6, pX4_3 RGFP966 n = 5, pX4_3 Entinostat n = 4, pX4_3 DMSO vs pX4_3 RGFP966 p = 0.0003, pX4_3 DMSO vs pX4_3 Entinostat p < 0.0003). **e** Immunofluorescence staining of human skin sections with Ctrl-IgG or pX4_3 and DSMO/ 5 μM RGFP966 or 10 μM Entinostat injection using KLF5 antibodies and DAPI. Representative images are shown. Scale bar = 100 μm. **f** Quantification of (**e**) displaying the mean of intensity of KLF5 per nucleus normalized to the average signal intensity within each experiment (Ctrl-IgG DMSO n = 6, pX4_3 DMSO n = 6, pX4_3 RGFP966 n = 5, pX4_3 Entinostat n = 4, Ctrl-IgG DMSO vs pX4_3 DMSO p < 0.0001, pX4_3 DMSO vs pX4_3 RGFP966 p = 0.0046, pX4_3 DMSO vs pX4_3 Entinostat p = 0.0012). **h** Schematic of the results. Values expressed as mean with standard deviation (mean ± SD). One n represents one biological replicate. Source data are provided as a Source Data file. All experiments were statistically analyzed with One-way-ANOVA, SIDAK correction. $p < 0.01^{**}$; $p < 0.001^{***}$.

## Regulatory mechanisms of DSG3 membrane localization and cell−cell adhesion

In addition to the HDAC3/KLF5 axis explored in detail, our screens revealed other potential regulators of desmosomal adhesion. According to gene ontology analysis, the 56 candidates were significantly enriched in biological processes such as "cellular membrane organization" and "vesicle-mediated transport" (Supplementary Fig. 1b). Several to our knowledge unknown potential regulators of desmosomes, such as LDL Receptor Related Protein 3 (LRP3), Golgi Associated PDZ And Coiled-Coil Motif Containing (GOPC) and the elastase ELANE were suggested by the screen and await further analysis. Besides these regulatory molecules, several hits have already been implicated in desmosome function. One example is the actin nucleating complex Arp2/3 and its component Actin Related Protein 2/3 Complex Subunit 4 (ARPC4)[29] and Merlin/Neurofibromin 2 (NF2)[30], a protein, which links the actin cytoskeleton with proteins in the cell membrane. Both proteins were highly ranked as positive regulators in our screen. Indeed, CK14-Cre-driven knockout of ARPC4 leads to a psoriasis-like phenotype and a downregulation of DSG1 and DSC1[29]. Vice versa, it was shown that DSG1 is required to promote Arp2/3-driven actin nucleation[31]. ARPC4 and Merlin may also localize other adhesion molecules such as DSG3 in the membrane. Moreover, DSG and DSC-containing vesicles are transported from the Golgi apparatus to the membrane to ensure the correct localization[32]. DSG1 is interacting with and recycled by the retromer endosomal trafficking component VPS35 to ensure epidermal differentiation and stratification[33]. In line with this report, we also detected VPS35 as positive regulator for DSG3 localization at the membrane suggesting a similar recycling mechanism (Supplementary Table 1). We further identified AP-2 complex related protein Adaptor Related Protein Complex 2 Subunit Sigma 1 (AP2S1), Adaptor Related Protein Complex 2 Subunit Alpha 1 (AP2A1) and Adaptor Related Protein Complex 3 Subunit Beta 1 (AP3B1), which may ensure DSG3 trafficking since they are endosome-associated proteins and implicated in clathrin-coated pit formation[34]. The identified DSG3-regulators also significantly cluster in "Gene expression" and include gene transcription co-activator Host Cell Factor C1 (HCFC1), the transcription initiator TATA-Box Binding Protein Associated Factor 6 Like (TAF6L) as well as KLF5. In general, KLF5 acts as a transcriptional activator and regulates proliferation and tissue morphogenesis. This is in line with data from the epidermis, where overexpression of KLF5 leads to abnormal differentiation and hyperkeratosis[35]. With regard to desmosomes, KLF5 was shown to regulate DSG1 and DSP in corneal epithelium[36] and DSG2 in the intestinal epithelium, where DSG3 is normally not expressed[37]. In our study in human keratinocytes, KLF5 was identified as direct binder to the DSG3 but not the DSG2 promoter, suggesting a tissue-specific regulation of individual desmoglein genes. Importantly, KLF5 levels were also reduced in PV patients and in PV-IgG-treated cells and epidermis. This strongly suggested an involvement in PV-IgG-mediated modulation of DSG3 levels which we explored further.

## Targeting HDAC3 and KLF5 as therapeutic approaches in PV

In PV, autoantibodies develop against DSG1 and DSG3 and result in blister formation in the skin and mucosa[8]. It is established that autoantibodies lead to altered turnover and stability of DSG3 and DSG1 through signaling-dependent and independent mechanisms, which results in loss of cell−cell adhesion and blistering[13]. Currently, PV patient treatment regimens rely on immunosuppression and/or B-cell depletion to reduce antibody production[38–40]. One option for targeted treatment of PV by supplementing or replacing coarse immunosuppression may be the targeted modulation of cell−cell adhesion. Here we exploited HDAC3- and KLF5-dependent regulation of DSG3 transcription, which was identified by our screens, to rescue DSG3 protein levels and membrane localization in pemphigus models. Indeed, forced expression of DSG3 in NHEKs has previously been shown to prevent PV-IgG-induced loss of cell−cell adhesion[41]. Interestingly, the rapid protective effect of glucocorticoids was shown to rely on the increase of DSG3 transcription via STAT3 inhibition[42]. Moreover, Rapamycin restored cell−cell adhesion in keratinocytes as well as in the passive transfer neonatal mouse model through the same mechanism[42,43]. Application of Rapamycin in patients, however, showed mixed results[44–46].

The application of HDAC3 inhibitors to enhance KLF5 stability, which we outlined in this study, may represent an alternative option to modulate DSG3 transcription. A weakness of this approach is the notion that both genetic HDAC3 deletion and overexpression resulted in impaired intercellular adhesion. This suggests that HDAC3 levels or activity need to be precisely calibrated to ensure intercellular adhesion. One reason for this behavior may be that, depending on the levels of HDAC3, different effector mechanisms are in place. For example, HDAC3 has functions requiring its catalytic domain while others rely on a scaffolding action[47,48]. Specifically, in the context of PV-IgG, it is possible that primarily the catalytical function is enhanced (as indicated by the increased activity in HDAC activity assays), whereas, in the long-term depletion context, the non-catalytic functions are also affected. Alternatively, long-term HDAC3 depletion may trigger compensatory mechanisms which itself may affect intercellular adhesion. At least in the context of the short-term inhibition by RGFP966 or Entinostat, we did not observe downregulation of DSG3 or loss of cell−cell adhesion under non-PV-IgG conditions, which might be in line with the notion that the inhibitors target the catalytic function of HDAC3[25]. Although these limitations may reduce the therapeutic window in which a pharmacologic targeting of HDAC3 is beneficial, the strong upregulation in the PV setting harbors the potential to return HDAC3 and (through KLF5) also DSG3 to normal levels.

While many silencing or knockout approaches evaluated the effects of DSG3 loss on cellular functions, the long-term results of increasing DSG3 levels are largely unexplored. It is possible that this may interfere with layer-specific patterning of DSG3 expression in the epidermis which might lead to altered differentiation. For example, it was shown that forced expression of DSG3 in suprabasal epidermal layers resulted in differentiation defects[49]. Moreover, it is known that

desmosomal cadherins, at least to some extent, are regulated in a compensatory manner. For example, depletion of DSG3 leads to upregulation of DSG2[50,51]. It is unclear whether such compensation effects are affected by long-term HDAC3 modulation and how this alters homeostasis in the setting of a differentiating epithelium. This, and the notion that HDAC3 inhibition affects the expression of a multitude of genes, will require careful dosing studies even though the skin is amenable to topical treatment.

Mechanistically, it was previously shown that HDAC1[17] and HDAC2[19] deacetylate and destabilize KLF5. We were unable to identify such a mechanism for HDAC3 as we did not see altered KLF5 acetylation in response to PV-IgG. KLF5 acetylation may still play a role in regulating adhesion as acetylation-deficient mutants failed to augment intercellular adhesion. However, our results favor a model in which signaling in response to PV-IgG (such as p38MAPK activation) induces HDAC3 levels, which in turn enhances binding to the KLF5 promotor, leading to histone deacetylation and reduced KLF5 transcription. This, in turn, results in impaired DSG3 levels which may contribute to reduced adhesion and blistering. Preventing HDAC3 activation releases the brake on the KLF5/DSG3 transcriptional regulation axis and results in enhanced intercellular adhesion. Along similar lines, Zhao et al. [52] observed that histone H3/H4 acetylation and histone H3K4/H3K27 methylation were significantly decreased in PBMCs from PV patients compared to healthy controls. Also, in pemphigus foliaceus (PF), epigenetic enzymes were also found to be significantly associated with the disease[53]. These results are in line with the possibility that epigenetic modifications in response to autoantibody binding contribute to the disease phenotype.

It is well established that a large array of pathways is modulated in PV patients and PV models in response to autoantibodies and contribute to disease. Among many others, PV-IgG induces overactivation of several kinases such as p38MAPK, ERK, Src, and EGFR, often in a fashion dependent on the antibody target (recently reviewed in refs. 54,55). Pharmacologic damping of these activated pathways was shown to prevent PV-IgG-induced effects in several in vitro and in vivo models[24,56–60]. Thus, modulating signaling molecules represents another option to stabilize desmosome function in PV. As a recent example, increasing cAMP levels, which promote cell–cell adhesion in a variety of tissues,[61] ameliorated PV-IgG-induced effects in vitro and in vivo[62]. It is interesting that, with the exception of EGFR, we did not observe these signaling molecules as significant hits on our screen. Most likely, targeting these essential pathways either leads to a growth disadvantage of the respective cells or is compensated by complementary pathways. In line with this explanation, the commonly applied signaling inhibitors mostly reduce the activity but not the levels of the respective molecule.

Together, by addressing HDAC3 and KLF5, we present an option to modulate intercellular adhesion both under homeostatic conditions as well as in the pemphigus disease setting. We believe that this and other targeted modulations of desmosome function should be explored in more detail and translated into clinical models to ultimately provide therapeutic approaches for patients.

## Methods

### HaCaT cell culture and generation of Cas9-expressing HaCaTs

Spontaneously immortalized HaCaT (Human adult high Calcium low Temperature) keratinocytes[63] were cultured in a humidified atmosphere of 5% $CO_2$ and 37 °C in Dulbecco's Modified Eagle Medium (DMEM) (Sigma-Aldrich, D6546) containing 1.8 mM $Ca^{2+}$ and complemented with 10% fetal bovine serum (Merck, S0615), 50 U/ml penicillin (VWR, A1837.0025, D6546), 50 μg/ml streptomycin (VWR, A1852.0100) and 4 mM L-glutamine (Sigma-Aldrich, G7513). HaCaT cells at confluency were treated with Ctrl-IgG or PV-IgG (1:100) purified as described[64]. IgG fractions were purified using affinity chromatography with protein A agarose (786-283, G-Biosciences),

eluted with 25 mM citrate buffer (pH 2.4), and immediately concentrated with Pierce Protein Concentrators (88515, Pierce). Thereby the buffer was exchanged for PBS. The titer of the PV-IgG was determined by ELISA. RGFP966 (Selleckchem, #S7229) or Entinostat (Selleckchem, #S1053) were dissolved DMSO to a concentration of 50 mM. The HaCaT cells were treated with different dilutions of the stock solution as specified in the figure legends. To generate a homogenous Cas9-expressing HaCaT cell line, cells at passage 30 were transduced with MOI 30 of lentivirus expressing LentiCas9-Blast (a gift from Feng Zhang[65]) and selected with 6 μg/mL blasticidine S hydrochloride (Sigma-Aldrich, #15205) for 6 days. These cells were used to generate single-cell-derived clones by plating 50 cells into 1× 96-well plate and the Cas9 editing efficiency was checked by transduction with lentivirus containing the reporter pKLV2-U6gRNA5(gGFP)-PGKBFP2AGFP-W (a gift from Kosuke Yusa[11]). This vector contains BFP, GFP, and a sgRNA targeting the GFP sequence. The cell clone with the highest editing efficiency (as measured by flow cytometry for the presence of BFP and loss of GFP) was selected to perform the Crispr screen (Supplementary Fig. 1A).

### Production and cell culture of NHEK cells

Foreskin tissue was obtained during circumcision from patients who gave written and informed consent in accordance with the local ethics committee. (Ethikkommission Nordwest- und Zentralschweiz-EKNZ; date of approval: 11.06.2018, project ID: 2018-00963). The skin samples were washed three times in PBS containing 300 U/mL of penicillin (#A1837, AppliChem), 300 U/mL of streptomycin sulfate (#A1852, AppliChem), and 7.5 μg/mL of amphotericin B (#A2942 Sigma-Aldrich). The skin was cut into 0.5 × 1 cm pieces after removing excess tissue, blood vessels, and parts of the dermis. For separation of dermis and epidermis, skin samples were immersed overnight at 4 °C in 5 mg/mL Dispase II solution (#D4693, Sigma-Aldrich) in HBSS (#H8264, Sigma-Aldrich) containing 300 U/mL penicillin, 300 U/mL streptomycin sulfate and 2. 5 μg/mL amphotericin B. The epidermis was detached, washed once in PBS, and digested in 0.25% trypsin and 1 mmol/L EDTA containing 100 U/mL penicillin and 100 U/mL streptomycin sulfate at 37 °C for 20 min. The activity was stopped by a 1:1 dilution with a 1 mg/ml solution of soybean trypsinic inhibitor (#10684033, Gibco) in PBS. Keratinocytes were isolated by scraping epidermal debris from the bottom of the dish and passing through a 70 μm cell filter (#431751, Corning, Somerville, USA). The isolated normal human epidermal keratinocytes (NHEK) were then seeded at a density of ~8 × 10⁴ cells/cm² in EpiLife medium containing 60 μmol/L $CaCl_2$ (#MEPI500CA, Gibco) and 1% human keratinocyte supplement (#S0015, Gibco), 1% Pen/Strep and 2.5 μg/mL amphotericin B. After 3 days, the medium was changed and amphotericin B was discontinued. The cells were seeded at 40,000/well into a 24-well plate and grown to confluency. Then 1.2 mmol/L $CaCl_2$ was added for 24 h to induce differentiation. Treatment of NHEK cells was performed similarly to HaCaT cells.

### Lentivirus generation and viral transduction

For lentivirus generation, HEK293T cells were co-transfected overnight with the packaging vector psPAX2, the envelope vector pMD2.G (both a gift from Didier Trono), and transfer vectors using TurboFect (Thermo Fisher Scientific, R0532). The medium was changed and the supernatant containing virus particles was collected and enriched using LentiConcentrator (OriGene Technologies, TR30025) after 48 h. HaCaT cells were transduced with lentivirus in the presence of 5 μg/mL polybrene (Sigma-Aldrich, TR-1003-G). Selection was initiated 24 h after virus withdrawal. A summary of the vectors applied is available in Supplementary Table 5.

### Crispr/Cas9 screen

The Cas9-expressing HaCaT clone was transduced with lentiviruses containing the Human Brunello CRISPR knockout pooled library

(Addgene, #73179, #73179-LV, #73178, #73178-LV) in the plasmid LentiGuide-Puro (a gift from David Root and John Doench[12]) at a MOI of 0.15 and coverage of approximately 400 cells/sgRNA. Cells were selected for 3 days with 1 µg/mL puromycin dihydrochloride (Thermo Fisher Scientific, BP2956100) and afterward, cells were expanded for 2–3 weeks while maintaining a coverage of 600 cells/sgRNA. On the day of sorting, $50 \times 10^6$ cells were collected as a control (termed pre-sort), and $60 \times 10^6$ cells were stained with Alexa 647 anti-human Dsg3 antibody (Santa Cruz Biotechnology, sc-53487 AF647) or with PE anti-human CD324 antibody (BD BioLegend, #324106) for fluorescence-activated cell sorting (FACS) with the BD Influx™ Cell Sorter (BD Biosciences). SYTOX™ Blue Dead Cell Stain (Thermo Fisher Scientific, #S34857) was used as a live-dead stain. Cells with the approximately 10–12% lowest and the 10–12% highest fluorescence intensity in the Alexa 647 or PE channel, respectively were isolated as the Dsg3$^{low}$ and the Dsg3$^{high}$ populations. The screen was performed in two biological replicates.

## Illumina library generation and sequencing

Genomic DNA was isolated from $50 \times 10^6$ cells prior to sorting and from 1.5 to $3 \times 10^6$ cells after cell sorting using the Blood & Cell Culture DNA Kit with QIAGEN Genomic-tip 500/G or 20/G, respectively (Qiagen, #13362, #13323) according to the manufacturer's manual. For Illumina library generation genomic stretches containing the sgRNA sequence were amplified using forward primers introducing a stagger 0–8 bases and reverse primers introducing an 8 base index (for primer sequences see Supplementary Table 5). For pre-sort samples 8 and for sorted samples 2–3 parallel reactions with 10 µg gDNA each and 1 reaction with 200 ng pooled plasmid DNA were performed using 1.5 µL Ex Taq Polymerase (Takara bio, # RR0001A), 0.2 mM dNTPs, 1× reaction buffer, 0.5 µM pooled forward primers and 0.5 µM reverse primer (Microsynth) per 100 µL reaction. Cycling conditions were as follows: $1 \times 95\,°C$ for 1 min; $28 \times 95\,°C$ for 30 s, 53 °C for 30 s, 72 °C for 30 s; $1 \times 72\,°C$ for 10 min. Parallel PCR reactions were pooled and 100–200 µL of each was purified using 0.8 SPRISelect reagent (Beckman Coulter, B23317) according to the manufacturer's recommendations. The quality and the concentrations of PCR amplicons were measured using the Agilent Bioanalyzer 2100. The pooled libraries at equal concentrations were sequenced with the NextSeq550 system (Illumina) using the HiSeq Rapid SBS Kit v2 with single-end reads of 55 bases in length and an 8 base index read.

## CRISPR screen analysis

Raw sequencing reads were trimmed using Cutadapt[66] with TTGTGGAAAGGACGAAACACCG as an adapter input with compulsory 22 overlap and allowing 2 indels or mismatches to remove the bases upstream of the sgRNA coding sequence including the variable stagger. Trimmed reads were aligned allowing for no mismatch with the human Brunello library using the count command of the MAGeCK v.0.5.9.2 software[67]. The resulting read count tables were analyzed using the test command of MAGeCK by comparing the two biological replicates of Dsg3$^{low}$ over Dsg3$^{high}$ or vice versa. Hits were identified within the highest-ranked genes that had a positive false discovery rate (FDR) < 0.2. Data presentation was done in R (version 4.0) including packages EnhancedVolcano[68], MAGeCKFlute[69]. Gene ontology analysis was performed with Cytoscape 3.10.0[70] and the application BiNGO 3.0.3[71]. The dataset is available (https://www.ncbi.nlm.nih.gov/geo/; GEO: GSE244919).

## Flow cytometry (FCM)

For FCM, HaCaT cells were harvested with Trypsin (Thermo Fisher Scientific, #T/3760/48) and transferred to a 96-well plate. Cells were washed with FACS buffer (PBS, 2% FCS, 0,1% Sodium azide) and incubated with anti-DSG3-647 (1:200; Santa Crus Biotechnology, #sc-53487) and Zombie Aqua (1:200, BioLegend, #423101) for 30 min

at 4 °C. Stained cells were washed with FACS buffer once, fixed with 4% Paraformaldehyde (in PBS, Fisher Scientific, #10131580) washed with FACS buffer and resuspended in FACS buffer. The staining of the cells was measured using the flow cytometry analyzer BC Cytoflex.

## RNA isolation and qRT-PCR

HaCaT cells were cultured in a 6-well plate and treated as indicated. For RNA isolation, HaCaT cells were harvested in 350 µl TriReagent (Lucerna Chem, #TR118). The RNA was isolated with a Direct-zol RNA MiniPrep Kit (Zymo Research, #301079) according to the manufacturer's protocol. The RNA was resuspended in 30 µl of water. Concentration was determined with a Nanodrop 1000 spectrometer (Witec AG). A maximum 1 µg of RNA was reverse transcribed using SuperScript™ III Reverse Transcriptase (Invitrogen, #300510) and Oligo-dT-Primers (Thermo Fisher Scientific, #300446) according to the manufacturer's protocol. The cDNA was diluted 1:50 in RNAse/DNAse-free water (Bioconcept, #301553) and 4.5 µl were used for master mix with Power SYBR Green Master Mix (2×, Thermo Fisher Scientific, #300427) including 0,5 µM primers (forward and reverse, Supplementary Table 5). Quantitative Real-time polymerase chain reaction (qRT-PCR) was performed using the StepOne Real-Time PCR System (Applied Biosystems, # 4376357).

## HDAC3 activity assay

To determine the HDAC activity in HaCaT cells, the Fluorometric Histone Deacetylase Activity Assay Kit (Abcam, #ab156064) One Step method was applied according to the manufacturer's protocol. Pellet of 100% confluent cells cultured in a T25 cell culture flask was dissolved in 50 µl HDAC activity buffer (50 mM Tris-HCl pH 8, 137 mM NaCl, 2.7 mM KCl, 1 mM MgCl2, 1% NP40). Protein concentration was determined using a BCA protein assay kit (VWR, #PIER23225), and 50–100 µg of protein extract was used for the assay. Trichostatin A (1 µM, included in Kit, Abcam, #ab156064) or RGFP966 (5 µM, Selleckchem, #S7229) was added at the beginning of the assay. Fluorescent product was measured every 2 min for 40 min, Ex 355/ Em 460 nm for 60 min. HDAC activity was calculated by the determination of the slope during the straight rise of the curve multiplied by the protein amount.

## Dispase-based dissociation assay

HaCaT cells were seeded in 1:2 dilution from a 100% T75 flask into 24 wells for at least 24 h until 100% confluency. The NHEK were seeded at 40,000/well into a 24-well plate and grown to confluency, followed by differentiation to allow them to spread. Then 1.2 mmol/L CaCl₂ was added for 24 h to induce differentiation. Then HaCaT or NHEK cells were washed with PBS and then incubated with 250 µl Dispase II (Sigma-Aldrich, #D4693) solution (50 mg in 10 ml HBSS) for 40 min in the dark at 37 °C. After incubation and detachment of the HaCaT monolayer, 150 µl of HBSS (Huber Lab, #A3140) were added. The monolayer was subjected to homogeneous shear stress by 10× electrical pipetting with Eppendorf Xplorer 1000 (350 µl, Eppendorf, #L49475G). The fragments were documented and quantified with a stereo microscope (Olympus, #SZX2-TR30) with an attached camera (Canon, #EOS 800D).

## Protein extraction and Western Blot

HaCaT or NHEK cells were washed once with PBS and then harvested in 100 µl SDS-lysis buffer 12.5 mM HEPES, 1 mM EDTA, 12.5 mM NaF, 0.5% SDS, 1× Protease Inhibitor Cocktail (Sigma-Aldrich, #11697498001). The protein amount was determined with a BCA protein assay kit (VWR, #PIER23225) according to the manufacturer's protocol and Western Blot was performed as described[10]. The following antibodies were used: KLF5 (1:1000, Active Motif, #61099), HDAC3 (Abcam, #ab7030), GAPDH (1:1000, Thermo Fisher Scientific, #16836913), H3ac

(1:1000, Lubioscience, #39040), DSG3 (Elab, #E-AB-62720), KLF5 K369ac (1:100, kindly provided by Jin-Tang Dong).

## Luciferase assay

The promoter fragment of DSG3 was cloned into pGL4.10 controlling the firefly luciferase (Supplementary Table 5). HaCaT cells were transfected with pGL4.10 DSG3 promoter or empty control and pGL4.73 containing Renilla luciferase controlled by an SV40 promoter as transfection efficiency control using TurboFect (Thermo Fisher Scientific, R0532) according to manufacturer's protocol. 24 h later cells were treated with human Ctrl-IgG (1:100) or Pemphigus vulgaris PV-IgG (1:100), with DMSO (Ctrl) or RGFP966 (Selleckchem, #S7229) with indicated concentrations and incubated for 24 h at 37 °C. The medium was aspirated and 35 µl of HaCaT medium was added. Then, 35 µl of Dual-Glo Reagent (Dual-Glo Luciferase Assay System, Promega, #E2920) was added, cells were homogenized via pipetting and 15 min incubated at room temperature. 50 µl of homogenate was subjected to a 96-well plate and measured with a microplate reader (BioTek, Synergy H1) for determination of the DSG3 promoter activity. Then 25 µl of Dual-Glo Stop and Glo reagent were added and incubated for 10 min, room temperature, and the Renilla luciferase signal was determined.

## HaCaT nuclear extract

For promoter pulldown, HaCaT nuclear extracts were prepared from 100% confluent T75 cell culture flasks. The cells were harvested via trypsination and after two washes with PBS, the cells were homogenized in 5 volumes of buffer A (10 mM Hepes KOH pH 7.6, 1.5 mM MgCl$_2$, 10 mM KCl) and incubated 10 min on ice, followed by centrifugation and removal of the supernatant. Then the cells were resuspended in 2 volumes of buffer A+ (A, 0.1% IGEPAL CA-630, 1× Complete protease inhibitors, 0.5 mM DTT) and Dounce homogenized for 50 strokes with a tight pestle. Cell suspension was centrifuged for 15 min at 1500 × $g$ and the supernatant was removed. Then, the cell pellet was washed in 10 volumes of ice-cold PBS and pelleted again (5 min, 3900 × $g$). The cells were resuspended in 2 volumes of Buffer C (420 mM NaCl, 20 mM Hepes KOH pH 7.6, 20% v/v glycerol, 2 mM MgCl$_2$, 0.2 mM EDTA, 0.1% IGEPAL CA-630, 1× Complete protease inhibitors, 0.5 mM DTT) and incubated for 1 h at 4 °C under constant rotation. The solution was then centrifuged for 14,000 × $g$ for 1 h and the supernatant was aliquoted and stored at −80 °C.

## Promoter pulldown

First, the DSG3 promoter sequence was amplified via PCR using Platinum Super Fi 2 DNA Polymerase (Thermo Fisher Scientific, #16410771) using primers described in Supplementary Table 5 (Annealing temperature: 58 °C) according to manufacturer's protocol. The forward primer is biotinylated at the 5' prime end. The PCR product was purified via phenol-chloroform extraction and the resulting pellet was dissolved in 400 µl of water. For 1 reaction 25 µg of biotinylated DNA was added to 125 µl of Streptavidin-coated Dynabeads (NEB, #S1420S) and incubated for 30 min, room temperature. After three times washing with PBS, 400 µg of nuclear extract in PD buffer (150 mM NaCl, 50 mM Tris/HCl pH: 8.0, 0.5% Igepal CA-630, 10 mM MgCl2, 1 mM DTT, Protease inhibitor cocktail) containing 10 mg/ml salmon sperm DNA (Invitrogen, #15632011) were added to the beads and incubated 90 min at 4 °C under constant rotation. Then the beads were washed 3 times with PD buffer and the beads were eluted with 1× Laemmli containing 100 mM DTT for 5 min at 95 °C.

## MS sample preparation and measurement

The MS measurement was done as detailed before[72]. In brief, protein samples were separated on a 4%–12% NuPAGE Novex Bis-Tris precast gel (Thermo Scientific) at 180 V in 1× NuPAGE MES buffer (Thermo Scientific) for 8 min. After protein fixation and Coomassie blue staining, individual lanes were cut, minced, and destained with 50% EtOH / 25 mM ammonium bicarbonate buffer pH 8.0 (ABC). The gel pieces were dehydrated with 100% acetonitrile (ACN) and afterward incubated with reduction buffer (10 mM DTT in 50 mM ABC) at 56 °C for 60 min followed by an alkylation step with iodoacetamide (50 mM IAA in 50 mM ABC) for 45 min. Gel pieces were completely dehydrated with ACN, dried, covered with trypsin solution (1 µg MS-grade trypsin per sample), and incubated overnight at 37 °C. Subsequently, peptides were extracted by incubation with extraction buffer (3% TFA and 30% ACN) for 15 min and dehydration with 100% ACN. ACN was completely removed in a concentrator (Eppendorf). Peptides were desalted on StageTips and separated on a with capillary (New Objective) packed with Reprosil C18 (Dr. Maisch) using an Easy nLC 1000 system (Thermo Scientific). Peptides were eluted from the column with a 90 min gradient from 2 to 60% ACN with 0.1% formic acid at a flow rate of 225 nL/min. The HPLC system was directly connected to a Q Exactive Plus mass spectrometer (Thermo Scientific). The mass spectrometer performed HCD fragmentation with a data-dependent Top10 MS/MS spectra acquisition scheme per MS full scan in the Orbitrap analyzer.

## MS data analysis and bioinformatics analysis

Raw files were processed with MaxQuant version 1.6.5.0 and searched against a human UNIPROT protein database using the Andromeda search engine. Standard MaxQuant instrument settings for the orbitrap were applied, i.e. Carbamidomethyl (Cys) was set as fixed modification, acetyl (N-term protein) and oxidation (Met) were considered as variable modifications, and trypsin (specific) was selected as enzyme specificity with maximal two miscleavages. LFQ quantitation and match between runs option were activated. The volcano plot was generated from a filtered MaxQuant proteinGroups output file with contaminants, reverse database hits, protein groups only identified by site, and protein groups with less than 2 peptides (at least one of them classified as unique) removed.

## Immunofluorescence staining and imaging

Cell cultures grown on coverslips were fixed with cold methanol for 10 min at 4 °C. Cells were then permeabilized with 0.1% Triton X-100 in PBS for 10 min and blocked with 3% BSA and 1% normal goat serum in PBS for 1 h. The coverslips were incubated overnight at 4 °C with the following primary antibodies: Mouse anti-Dsg3-mAb (Invitrogen, #326300), rabbit anti-KLF5 rabbit (Active Motif, #61099), anti-DSP-mAb (NW39), HDAC3 (Sigma, #HPA052052). Cells were then washed three times with PBS and incubated with the secondary AlexaFluor-coupled antibodies (Fisher Scientific, A-11008, A-11004) for one hour at room temperature. DAPI (Sigma-Aldrich, D9542) was added for 10 min. The coverslips were washed three more times with PBS and fixed with ProLong Diamond Antifade (Thermo Fisher Scientific, P36961). Images were taken with an HC PL APO CS2 63×/1.40 oil objective on a Stellaris 8 Falcon confocal microscope (Leica, Wetzlar, Germany).

## Immunoprecipitation

Cells were harvested in IP buffer (50 mM Tris-HCl pH 8, 170 mM NaCl, 0.1% NP40, 5% glycerol, cOmplete Roche: 1 tablet/50 mL) and lysed by repeated thawing and freezing cycles. Debris was removed by centrifugation at full speed (17,000 × $g$ in a bench-top centrifuge) for 5 min at 4 °C and harvesting of the supernatant. The amount of protein was determined by BCA. Equal amounts of protein were used for control IP (4 µg normal rabbit IgG, Cell Signaling 2729S) and IP of KLF5 (4 µg, KLF5 61099 rabbit pAb, Active Motif) and incubated overnight at 4 °C with constant shaking (20 rpm). The next day, Protein G coated beads were added to the lysates (30 µl, Dynabeads Protein G, Invitrogen 10003D, washed three times with IP buffer) and incubated for 1 h at 4 °C. After incubation the beads were then washed three times for 10 min with IP buffer under constant rotation and eluted with 1× Laemmli.

## Chromatin-Immunoprecipitation

ChIP-qPCR was performed as detailed before[73]. Confluent HaCaT cells were crosslinked with 1% formaldehyde for 20 min at 4 °C. After washing with PBS, cells were harvested in lysis buffer (50 mM Tris pH 8.0, 1 mM EDTA, 1% SDS, and 1× protease inhibitor cocktail [11873580001, Roche]) and sonicated using a Bioruptor (Diagenode) to achieve DNA fragments of 200−500 bp. For each ChIP experiment, 1 µg of the following antibodies were used per ChIP: anti-HDAC3 (Abcam, ab137704) and anti-KLF5 (Active Motif, 61099) antibody was used. Following washing, the immunoprecipitated DNA was eluted with a buffer containing 0.1 M sodium bicarbonate and 1% SDS for 1 h at room temperature. The crosslinks were reversed by incubating at 65 °C overnight. Finally, the DNA was purified using Nucleospin gel and pcr clean up kit (740609.50S, Macherey-Nagel) and analyzed in a 1:10 dilution with quantitative Real-time PCR.

## Passive transfer neonatal mouse model and ex vivo skin and mucosa models

The mouse experiments have been approved by Veterinäramt Basel-Stadt (Number 3159). The animal experiments were performed to the ARRIVE guidelines. For bedding, we use LTE E001 L10 Classic Chips from ABEDD SIA. In addition to the bedding, we provide 2–4 Kleenex tissues as nesting material to enhance the animals' comfort. To offer additional enrichment, we sometimes place half-empty Kleenex boxes in the cages. For animals housed individually, we ensure they have a running wheel in their cage to promote physical activity and mental stimulation. The dark/light cycle was maintained at 12 h of light and 12 h of darkness (6 am–6 pm). The temperature was kept at 22 ± 2 °C, and the relative humidity was maintained between 45% and 65%. Newborn BALB/c mice at p1 received an intracutaneous injection of 40 µg pX4_3 (purified according to published protocols[16]) or 40 µg control IgG in PBS as a control containing 5 µM RGFP966 in DMSO or DMSO only using a 29G needle. One paw was then inked using a tattoo kit (Ketchum, XKA-2300087) and neonates were returned to the cage. In this way, it was possible to distinguish up to 4 groups. 24 h after the last injection, the animals were euthanized by decapitation and the skin was prepared for histological examination.

The human ex vivo pemphigus model was conducted using cadavers from the Human Body Donor Program without a history of skin disease from the Department of Biomedicine, University of Basel, Switzerland. Experiments and protocols are covered by the Body Donation Programme of the Department of Biomedicine, University of Basel, Switzerland. Written informed consent for the use of research samples was obtained from the body donors. For the ex vivo skin or mucosa model, human skin pieces (approx 1 cm$^2$) were collected from the thigh or the inner lip of body donors without history of skin disease, which arrived at the Anatomical Institute of the University of Basel within 24 h after decease. Tissues at this time and after additional 24 h of ex vivo incubation are still viable[27]. Written informed consent was obtained from body donors for use of tissue samples in research. Tissue pieces were injected superficially with 40 µg of pX4_3 or 40 µg of control IgG in PBS. The specimens were incubated floating on DMEM (Sigma, D6546) including 10% FCS, 0.2% Glutamate and 0.5% Penicillin-Streptomycin for 24 h. After the application of a shear stress (10 × 90° rotations with a custom-made rubber-stencil), the skin was subjected to histological examination.

## Pemphigus vulgaris patient material

Pemphigus vulgaris patient material was obtained from patients who gave written and informed consent in accordance with the local ethics committee (approved by the Ethics Commission of the Medical Faculty at the University of Marburg under the number 169/19). Skin or mucosa punch biopsies were taken from Pemphigus vulgaris patients (ages between 34 and 97) as part of the diagnostic procedures by the Department of Dermatology, Universitätsklinikum Giessen Marburg

(UKGM) (Supplementary Table 4). The control sections originate from excess normal tissue removed during the resection of skin tumors. Biopsies were paraffin-embedded according to standard procedures. Sections were deparaffinized as described below and stained with anti-KLF5 antibodies and DAPI.

## Histology and immunostaining of tissue sections

Human or mouse tissue was fixed with 4% PFA in PBS at 4 °C for 24 h, washed three times with PBS and embedded in paraffin according to standard procedures using the TPC 15 Tissue Processor (Medite Medizintechnik). Paraffin blocks were cut into 5 µm thick sections using a microtome (Thermo Fisher Scientific, HM355S). Hematoxylin and eosin (H&E) staining was carried out according to standard procedures. Briefly, the sections were stained for 5 min with Mayer's hemalaun solution (Sigma-Aldrich, No. 1.09249.1022), rinsed, subjected to dehydration in an increasing ethanol series and counterstained for 5 min with 0.5% (w/v) eosin solution. Following washing steps in ethanol and methyl salicylate, the sections were covered with DPX mounting medium (Sigma-Aldrich, #06522). For immunostaining, the sections were deparaffinized and antigen retrieval was performed in citrate buffer (10 mM citric acid monohydrate (20276.235, VWR), pH 6, 0.1% Triton) for 20 min at 95 °C. The material was permeabilized in 0.1% Triton X-100 in PBS for 5 min and blocked with 3% bovine serum albumin/0.12% normal goat serum in PBS for 1 h. The sections were incubated with the anti-KLF5 antibody (Active Motif, #61099), anti-HDAC3 antibody (Sigma-Aldrich, #HPA052052) or anti-DSG3 antibody (Progen, #65193) in PBS overnight at 4 °C. After washing, the secondary antibodies were added for 1 h, RT. DAPI (Sigma-Aldrich, D9542) was added for 10 min to stain the nuclei. Samples were embedded with Fluoromount Aqueous Mounting Medium (Sigma-Aldrich, F4680).

## Statistics and image analysis

Statistical analysis was carried out using GraphPad Prism 8. Data sets were first tested for normal distribution using the Shapiro-Wilk normality test. When appropriate, one-sample t-test was used to compare data sets with theoretical values of 1 or 0 (when the Ctrl value was normalized to 1 or 0), students t-tests to compare two data sets and one-way ANOVA corrected with Sidak's multiple comparisons test was performed for more than two data sets to determine statistical significance (p < 0.05). Where reasonable, the Welch correction for unequal variances was applied. Error bars in all graphs are presented as ±SD. For each experiment, at least 3 biological replicates were used. The figures were created with the use of Photoshop CC and Illustrator CC (Adobe, San José, CA, USA). Immunofluorescence staining of cells was analyzed with with ImageJ software for quantifying DSG3 mean intensity in the cell membrane by drawing an ROI around the plasma membrane. The DSG3 intensity was then normalized by the respective length of plasma membrane. Immunofluorescence stainings of tissue sections were analyzed with QuPath-0.4.3. Cell nuclei were detected using DAPI, then the intensity of the protein of interest was determined for each nucleus. The mean intensity of the staining was divided by the number of nuclei. H&E and immunostained samples were scanned with a slide scanner (NanoZoomerS60, Hamamatsu). Blister and total length of each H&E-stained sample was measured with NDPview.2.

## Reporting summary

Further information on research design is available in the Nature Portfolio Reporting Summary linked to this article.

## Data availability

The CRISPR screen data generated in this study have been deposited in the Gene Expression Omnibus database under accession code GSE244919 URL. The KLF5 ChIP sequencing dataset is available under GEO: GSE168600. The HDAC3 ChIP sequencing dataset is available under GEO: GSE137232. All data supporting the findings of this study

are available within the paper and its Supplementary Information. Source data are provided with this paper.

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

## Acknowledgements

The authors thank Dr. Diego Calabrese (Histology Core Facility), Dr. Mike Abanto, and Dr. P. Lorentz (Microscopy Core Facility), Dr. Florian Geier (Bioinformatics Core Facility), and Mireille Toranelli, Department of Biomedicine, University of Basel, Switzerland. We are grateful to Dr Katarzyna Buczak, Proteomics Facility Basel (Biozentrum, University of Basel). We thank Prof. Kathleen Green, (Northwestern University, USA) for the provision of desmoplakin antibodies and the Department of Urology (University Hospital Basel) for providing human preputial tissue. We are grateful to Dr. Jin-Tang Dong (SUSTech School of Medicine, China) for providing the KLF5ac antibody. The experiments were supported by the Swiss National Science Foundation (#197764) and a grant from the Novartis Foundation for Biomedical Research (both to VS).

## Author contributions

H.F., M.R., A.Z., C.S., V.B., K.L.F., P.H., M.S., and F.B. performed the experiments and collected and analyzed the data. T.C., D.D., M.H., and E.S. provided reagents and patient material. H.F. and V.S. designed the study. H.F. and V.S. drafted the manuscript. All authors critically reviewed the manuscript.

## Funding

## Competing interests

The authors declare no competing interests.
