## [Peer Review File · Nature Communications]

REVIEWER COMMENTS

Reviewer #1 (Remarks to the Author):

The authors attempt to elucidate the mechanisms controlling desmosomal adhesion that underpin the pathogenicity of autoantibodies developed by patients with Pemphigus vulgaris and hence inform potential treatments.

While the authors provide an extensive set of experiments to delve into these topics, there are two important flaws that hamper my enthusiasm with this study.

Firstly, key logical passages reconciling the role of different molecules highlighted in this study rely on assumptions or literature findings and were not experimentally verified (Comments 1 to 4). Secondly, the choice of cell lines may be itself biased and may need additional verification on several more keratinocyte strains (Comments 5, 6). Finally, there are key technical and experimental queries that need clarification (comments 7 to 11).

1. "Because interference with the stability of the desmosomal adhesion molecule DSG3 at the cell membrane leads to loss of intercellular adhesion in disease"

This is an overstatement and undermines some of the work undertaken in this study. Specifically, Dsg3 null mice do not show gross defects of cell-cell adhesion and Dsg3^{-/-} cells are microscopically indistinguishable from WT (Nguyen et al., JCI 2000).

2. "We chose to evaluate membrane levels of DSG3 and use these as proxy for intercellular adhesion." The flaw with this reasoning is that Dsg3 is not essential for cell-cell adhesion, at least generally speaking. There are so many ways to sort cells with divergent adhesion properties, so I'm not sure why the authors selected the expression levels of membrane Dsg3 for such an important aim.

It would also be essential to establish the baseline expression levels of other adhesion molecules (particularly Dsg1) or to run the screening when HaCaT are in their hyper-adhesive state – which most likely replicates a calcium-independent adhesive state present in the skin.

3. "The promoter-bound protein fraction was then analyzed by quantitative mass spectrometry". The authors need to confirm this through a reverse approach, i.e. ChIP with anti-KLF5 Abs.

4. "It is established that the activity of KLF transcription factors is regulated by acetylation and through histone deacetylases (HDACs, Supplementary Fig. 3d)."

This is perhaps the most important statement for a mechanistic understanding of PV IgG pathogenicity. Unfortunately, the authors fail to demonstrate this experimentally. So, the logical passages between “unbiased” screening, KLF5 involvement, and HDAC3 activation is based on literature and was not validated. The authors need to design an experiment where Dsg3, KLF5, and HDAC3 and linked to PV IgG activity in the same experiment using appropriate inhibitors and activators and where their sequential interactions and cause-effect is shown downstream of PV IgG action.

5. “A whole-genome CRISPR screen (Fig. 1a) using the human HaCaT keratinocyte cell line.” Whilst HaCaT cells form normal-looking skin in vivo and are a widely used model of normal keratinocytes, they are still cell lines, are immortal and feature mutations that affect their proliferation, aging and genetic stability. This is particularly important for Dsg3 regulation as its expression is affected by cell cycle-related pathways. The “unbiased” screening should have been replicated with NHEKs and possibly mucosal keratinocytes. The latter are most relevant because these are thought to express lower levels of Dsg1, and hence Dsg3 is more important for cell-cell adhesion in mucosal keratinocytes.

6. HaCaT keratinocytes are skin cells. Yet when PV patients develop anti-Dsg3 IgG (in the absence of autoAbs against Dsg1 and/or other epithelial targets), no skin blisters occur. Hence, it is difficult to see the logic on focusing on Dsg3 as the key adhesion molecule in skin keratinocytes when in fact it is not clinically relevant in cutaneous PV blistering. In their initial unbiased screening, the authors should at the very least use few other adhesion molecules to sort/stratify their cells and see what results overlap.

7. Have the authors considered to silence or knock down DSG3 to see if the same changes to KLF5 and HDAC3 occur?

8. “To investigate if HDAC3 inhibition reverses PV-IgG-induced loss of cell-cell adhesion...” I’m not entirely clear how did the authors assess this – did they treat cells with PV IgG and inhibitor concurrently? (in which case the inhibitor prevents, rather than reverse, acantholysis). Or the inhibitor was added later?

9. “eight PV patients versus eight healthy control skins” can you provide details of the clinical and immunological profile of these patients.

10. “using PV-IgG and px4_3” can you characterise the immunophenotype of the PV IgG please

11. Figure 2d. why is HDAC3 activity going up over time in controls?

Reviewer #2 (Remarks to the Author):

Franz et al performed a genome-wide Crispr/Cas9 screen in human keratinocytes to identify putative regulators of cell-cell adhesion using DSG3 expression as a readout. Moreover, the authors performed a Dsg3 promoter pull-down combined with MS approaches to identify Dsg3 promoter binding proteins. As outcome of both approaches, the authors identified KLF5 as a positive regulator of DSG3 expression and intercellular adhesion. Lowering KLF5 expression resulted in a diminished cell-cell adhesion of HaCaT cells, while enforced expression of KLF5 enhanced cell-cell adhesion. They further observed that KLF5 protein levels were reduced in skin biopsies of pemphigus vulgaris patient. Incubation of cell lines with PV-IgG autoantibodies resulted in the reduction of KLF5 expression and a parallel upregulation of the expression of HDAC3, which is a known KLF5 interactor, and in an increase in HDAC activity. The authors also observed acetylation of KLF5 in HaCaT cells. Furthermore, while pan HDAC inhibitors severely blocked HDAC activity in HaCaT cells, an HDAC3-specific inhibitor ameliorated the PV-IgG induced increased in HDAC activity, suggesting that HDAC3 is the primary HDAC family member activated in response to antibody treatment. Overexpression of HDAC3 reduced cell-cell adhesion of HaCaT cells, similar to sgRNA-mediated HDAC3 down-regulation, suggesting that HDAC3 expression levels have to be tightly controlled for the proper regulation of cell-cell adhesion. Finally, HDAC3 inhibitor treatment attenuates PV-IgG induced blister formation in mouse and human skin. Based on these data the authors conclude that KLF5 and HDAC3 are novel regulators of DSG3 gene expression and hence cell-cell adhesion. Overall, this is an interesting and nicely performed study. I have the following comments:

(1) The authors showed that KLF5 is acetylated in HaCaT cells (SupFig.3d). Is the acetylation status regulated by HDAC3? Does overexpression of HDAC3 reduces KLF5 acetylation? And vice versa, does the application of an HDAC3 inhibitor increases KLF5 acetylation?

(2) Related to KLF5 acetylation status: lysine residue (K369) that are targeted by acetylation within KLF5 have been previously identified (e.g. PMID: 33731701). What happens if one overexpresses acetylation-deficient and acetylation-mimicking mutant forms (K to R and K to Q mutants, respectively)? Does overexpression of a mutant KLF5 still enhances DSG3 expression or even result in a higher expression compared to the effect from WT KLF5?

(3) Is the observed drop in KLF5 protein levels induced by PV-IgG treatment caused by reduced Klf5 gene expression or by posttranslational (or post transcriptional) changes? The authors should determine Klf5 gene expression levels in response to PV-IgG treatment. Moreover, to address KLF5 protein stability, the

authors should perform cycloheximide experiments to study whether the half-life of KLF5 is changed in responses to PV-IgG treatment.

(4) Overexpression of HDAC3 resulted in the down-regulation of DSG3 (Fig. 3f), and down-regulation of HDAC3 also resulted in reduced DSG3 protein expression (SupFig. 3g). How can this be explained? HDAC3 also interacts with other factors as well as will class II HDACs. Could one explanation be that catalytic and non-catalytic functions (i.e. a scaffolding role) for HDAC3 depend on expression levels? Some enzymatic inactive forms of HDAC3 has been described (e.g. PMID: 24268577). What happens if one overexpresses a catalytically-dead HDAC3 version? Does this lead to a down-regulation of DSG3? And what happens if HDAC3 is down-regulated (instead of using an HDAC3 inhibitor) and cells are treated with PV-IgG?

(5) It is not clear, at least to this reviewer, why PV-IgG treatment results in the downregulation of KLF5 as shown in Figure 3c. Is there a signaling process induced upon binding of anti-DSG3 antibodies (PV-IgG)? If yes, does this mean that there is a feedback loop that result in the downregulation of KLF5 protein and hence a further decrease in DSG3? Since the authors observed that HDAC3 is upregulated upon PV-IgG binding (Figure 3e), do they propose a model in which PV-IgG treatment results in an upregulation of HDAC3 followed by a deacetylation of KLF5 and as a consequence reduced KLF5 stability? This is also related to my comment #1. The authors should consider to draw a model of how this might work.

Reviewer #3 (Remarks to the Author):

Here, Franz et al identify KLF5 and HDAC3 as regulators of DSG3 protein level and cell adhesion, and positions them as possible therapeutic targets in PV. The topic is of interest to the field, the manuscript is extremely clear and well written, and the experiments well designed and clearly presented. As such this work represents an important contribution to the field. Several points below I believe would benefit from further discussion and exploration. These are not major and overall I find this work highly compelling.

Line 164-168 “However, transduction of HaCaT cells with sgRNA targeting HDAC3 also resulted in a significant reduction of DSG3 protein levels (Supplementary Fig 3g). This was accompanied by a significant decrease of cell-cell adhesion” . The authors note that this suggests precisely balanced activity, this notion deserves more discussion or exploration. I was comparing this with the pharmacological inhibition of HDAC3, which appears to show no significant change in adhesion or Dsg3 protein (Fig 4 a-b and Sup. Fig 4). This suggests something distinct between these two modes of suppressing HDAC3 which I would like the authors to discuss.

In the mouse or human explant are there changes in Dsg3 protein level following PV which are rescued by inhibiting HDAC3? This result would be expected based on the cell culture work, however demonstration in the PV in vivo and ex vivo models would strengthen the mechanistic claims of the manuscript.

Another interesting question came to mind when reading this experiment. It is not clear if there would be different changes based on the epidermal layer - ie will inhibition of HDAC3 lead to expression of Dsg3 in the supra-basal layers of the epidermis? This could be tested, or added to the discussion of the manuscript.

Finally, it is interesting to consider the roles of other desmosomal proteins. If Dsg3 is down regulated, are other cadherins up-regulated? The authors note in the discussion that KLF5 has been shown to regulate DSP, DSG1, and DSG2 in other tissues. Are these proteins also being regulated in keratinocytes upon inhibition of HDAC3?

This is beyond the scope of the work, and I am not requesting an experiment, only discussion. Would treatment with an HDAC3 inhibitor could rescue an adhesion defect? Would there be an issue with altering DSG3 protein levels in healthy skin? Impacts on stratification? I appreciate this may be difficult to speculate on, but I found these questions exciting.

Minor

In Figure 3D there are two panels, the figure legend does not describe the left panel (RFU vs time).

We thank all reviewers for their thorough and constructive feedback. Please find the responses underneath the respective remarks.

Reviewer #1 (Remarks to the Author):

The authors attempt to elucidate the mechanisms controlling desmosomal adhesion that underpin the pathogenicity of autoantibodies developed by patients with Pemphigus vulgaris and hence inform potential treatments.

While the authors provide an extensive set of experiments to delve into these topics, there are two important flaws that hamper my enthusiasm with this study.

Firstly, key logical passages reconciling the role of different molecules highlighted in this study rely on assumptions or literature findings and were not experimentally verified (Comments 1 to 4). Secondly, the choice of cell lines may be itself biased and may need additional verification on several more keratinocyte strains (Comments 5, 6). Finally, there are key technical and experimental queries that need clarification (comments 7 to 11).

We thank the Reviewer for his profound review and detailed suggestions. We have addressed the topics both experimentally as well as in the discussion. For a detailed reply please see below.

1. "Because interference with the stability of the desmosomal adhesion molecule DSG3 at the cell membrane leads to loss of intercellular adhesion in disease" This is an overstatement and undermines some of the work undertaken in this study. Specifically, Dsg3 null mice do not show gross defects of cell-cell adhesion and Dsg3^{-/-} cells are microscopically indistinguishable from WT (Nguyen et al., JCI 2000).

We agree with the reviewer that loss of DSG3, depending on the cellular or tissue context, can be at least partially compensated by other desmosomal and non-desmosomal adhesion molecules. Still, DSG3 is a central antigen in Pemphigus vulgaris and is required for strong intercellular adhesion. As examples, (i) adult DSG3 knockout mice show epidermal splitting (Rötzer et al. PMID 26763450), (ii) adult mice injected with AK23, a monoclonal DSG3 antibody derived from a pemphigus mouse model, develop splits within the hair follicles (Hariton et al., PMID 29105150), (iii) AK23 injection alone induces blistering in neonatal mice (Spindler et al., PMID 23298835), (iv) CRISPR/Cas9-mediated knockout of DSG3 leads to loss of adhesion in HaCaT keratinocytes (31178865), (v) a considerable amount of patients present with epidermal lesions despite being positive for anti-DSG3 antibodies only (e.g., Sielski et al., PMID 36211362). To better reflect the situation, we adapted the sentence mentioned by the reviewer to "Because interference with the amount and stability of desmosomal adhesion molecules at the cell membrane of epidermal keratinocytes or mucosal epithelial cells leads to loss of intercellular adhesion in pemphigus [...]"

2. "We chose to evaluate membrane levels of DSG3 and use these as proxy for intercellular adhesion." The flaw with this reasoning is that Dsg3 is not essential for cell-cell adhesion, at least generally speaking. There are so many ways to sort cells with divergent adhesion properties, so I'm not sure why the authors selected the expression levels of membrane Dsg3 for such an important aim.

It would also be essential to establish the baseline expression levels of other adhesion molecules (particularly Dsg1) or to run the screening when HaCaT are in their hyper-adhesive state – which most likely replicates a calcium-independent adhesive state present in the skin.

To be able to perform the whole genome CRISPR screening approach, there are several critical parameters that need to be matched with regard to the model system. First, a cell culture system which is reliably growing without donor variability is required, second, the cells are available in large quantities, third, each cell has a similar copy number and expression level of genetically introduced Cas9, and fourth, the activity of Cas9 is highly similar throughout all cells. This rules out basically all primary cells, some cellular states (e.g. hyperadhesion, at least during transduction and growth of the library) and many approaches to functionally sort for adhesive properties.

We chose DSG3 as proxy due to the reasons outlined in Comment 1. Moreover, in our settings applied here, deleting DSG3 by CRISPR/Cas9 gene editing resulted in a clear reduction of intercellular adhesion as seen in the dissociation assays (Fig. S1D), showing that in the conditions used, DSG3 is clearly required for full intercellular adhesion. HaCaT cells do not express DSG1 under normal cell culture conditions (see, for example, Spindler et al., PMID 21864491).

We agree that other approaches - which are however not applicable to this screen - might be more ideally suited to quantify cell adhesion. To overcome this situation, we carefully verified for the top hits in an unrelated clone of HaCaT cells that i) loss of the target gene, ii) loss of DSG3 membrane localization and iii) loss of cell-cell adhesion correlate.

3. “The promotor-bound protein fraction was then analyzed by quantitative mass spectrometry”. The authors need to confirm this through a reverse approach, i.e. ChIP with anti-KLF5 Abs.

We fully agree with the reviewer and performed chromatin immunoprecipitation using KLF5 antibodies. We then quantified the immunoprecipitated DNA with five different DSG3 primers using real-time PCR. We were able to localize KLF5 in the vicinity of the transcriptional start site (TSS) (Supplemental Fig. 2c). As a positive control, we detected KLF5 at the CTNNB1 promoter, which was shown to be regulated by KLF5 (Tang et al., PMID: 29760406) (Supplementary Fig. 2c). ChIP-sequencing data of KLF5 in HaCaT cells (GSM5150531) further corroborate these data (Supplementary Fig. 2d).

4. “It is established that the activity of KLF transcription factors is regulated by acetylation and through histone deacetylases (HDACs, Fig. 3d).”

This is perhaps the most important statement for a mechanistic understanding of PV IgG pathogenicity. Unfortunately, the authors fail to demonstrate this experimentally. So, the logical passages between “unbiased” screening, KLF5 involvement, and HDAC3 activation is based on literature and was not validated. The authors need to design an experiment where Dsg3, KLF5, and HDAC3 and linked to PV IgG activity in the same experiment using appropriate inhibitors and activators and where their sequential interactions and cause-effect is shown downstream of PV IgG action.

We agree with the reviewer that a better understanding of the mechanistic interplay of PV-IgG, HDAC3, KLF5 and adhesion is required. We added several new experiments to provide further data:

- i) *To further investigate the acetylation status of KLF5 (Supplementary Fig. 4f), we treated HaCaT cells with IgG or PV-IgG. By Western blotting using specific KLF5*

acetylation antibodies (Fig. 4a) we did not detect changes in the fraction of acetylated KLF5. Moreover, we did not detect KLF5 acetylation reliably by proteomic analysis (data not included), which might be a result of technical limitations.

- ii) In parallel to i), we generated acetylation-deficient mutants of KLF5 and tested these together with PV-IgG. While overexpression of wildtype KLF5 increased intercellular adhesion under basal conditions and in response to PV-IgG, the two acetylation-deficient mutants failed to do so (Fig 3d, and Supplementary Fig. 6a).
- iii) From i) and ii) we concluded that, although KLF5 acetylation promotes intercellular adhesion, PV-IgG do not alter KLF5 acetylation states. However, we noted that KLF5 mRNA was reduced upon PV-IgG treatment and increased in response to HDAC3 depletion, suggesting a regulatory mechanism during transcription of the gene (Fig. 4b). Indeed, we detected HDAC3 to be bound to the KLF5 promotor, which was strongly increased upon PV-IgG treatment under conditions in which also HDAC3 protein levels were upregulated (Fig. 4f-g). This suggests that HDAC3 represses KLF5 transcription by binding to the KLF5 promotor in response to PV-IgG.
- iv) We further show that the p38MAPK inhibitor SB202190 prevented HDAC3 increase and DSG3 loss in response to PV-IgG, linking a well-established pathway in pemphigus with HDAC3/KLF5 regulation (Fig. 4h).
- v) Finally, we combined PV-IgG treatment, HDAC3 inhibition and genetic loss of KLF5 in one experiment. While PV-IgG-induced DSG3 depletion and loss of adhesion were prevented by HDAC3 inhibition in sgNT cells, this effect was not detectable in KLF5-KO cells (Fig. 7e and 7g).

Together, our data are in agreement with a model in which PV-IgG led to an upregulation of HDAC3 in a p38MAPK-dependent manner. This in turn reduces KLF5 expression, which results in reduced DSG3 protein levels and impaired intercellular adhesion.

5. "A whole-genome CRISPR screen (Fig. 1a) using the human HaCaT keratinocyte cell line." Whilst HaCaT cells form normal-looking skin in vivo and are a widely used model of normal keratinocytes, they are still cell lines, are immortal and feature mutations that affect their proliferation, aging and genetic stability. This is particularly important for Dsg3 regulation as its expression is affected by cell cycle-related pathways. The "unbiased" screening should have been replicated with NHEKs and possibly mucosal keratinocytes. The latter are most relevant because these are thought to express lower levels of Dsg1, and hence Dsg3 is more important for cell-cell adhesion in mucosal keratinocytes.

As outlined in detail in comment 2, we have to rely on well-established cell culture models to perform the screen. It is clear that the results of a high throughput screen, which in most cases show limitations, need to be carefully verified in other suitable model systems. Accordingly, we validated the results from the screen in independent HaCaT clones and performed key experiments in NHEK cells. Moreover, we now individually checked patient epidermis and mucosa for HDAC3 and KLF5 levels, showing similar results as observed in cell culture systems (Fig. 3 a-c; 3f-h). Finally, we now included an ex vivo culture system of human mucosa in addition to murine in vivo and human skin ex vivo models, showing that blister length was significantly reduced by treatment with HDAC3 inhibitors (Supplementary Fig. 10c). These validation experiments from several cell culture systems, patient tissue and in vivo and ex vivo approaches in our view demonstrate the high applicability of our findings from the screen.

6. HaCaT keratinocytes are skin cells. Yet when PV patients develop anti-Dsg3 IgG (in the absence of autoAbs against Dsg1 and/or other epithelial targets), no skin blisters occur. Hence, it is difficult to see the logic on focusing on Dsg3 as the key adhesion molecule in skin keratinocytes when in fact it is not clinically relevant in cutaneous PV blistering. In their initial unbiased screening, the authors should at the very least use few other adhesion molecules to sort/stratify their cells and see what results overlap.

We agree with the reviewer that using DSG3 in skin keratinocytes is a compromise. We have explained in comments 1, 2 and 5 why we still chose this model system and molecule. In our view, we have performed a wide range of validation experiments to show that the results are applicable to the in vivo situation, specifically also to the mucosa. Indeed, including additional molecules would be highly interesting, but it would also mean that the entire screen would have to be replicated individually for each molecule of interest. This would mean extensive testing of detection antibodies suitable for FACS sorting, large costs for the library and sequencing and extensive bioinformatical analysis. We hope that the reviewer agrees that these experiments are beyond the time limits and scope of this revision.

7. Have the authors considered to silence or knock down DSG3 to see if the same changes to KLF5 and HDAC3 occur?

Thank you for this suggestion. We performed Western blot analysis of HaCaT cell lysates comparing DSG3-depleted cells to sgNT controls. In this experiment, depletion of DSG3 did not result in significant changes in KLF5 or HDAC3 (Supplementary Fig. 6c). This demonstrates that HDAC3/KLF5 is situated upstream of DSG3 transcription.

8. “To investigate if HDAC3 inhibition reverses PV-IgG-induced loss of cell-cell adhesion...” I’m not entirely clear how did the authors assess this – did they treat cells with PV IgG and inhibitor concurrently? (in which case the inhibitor prevents, rather than reverse, acantholysis). Or the inhibitor was added later?

We apologize for the misleading statement, in which “reverses” is now replaced with “prevents”. However, in addition to co-incubation in vitro or co-injection of the HDAC3 inhibitors into mouse skin, human skin (and newly into human mucosa), we now also included an experiment in which human skin was injected with HDAC3 inhibitors 2 hours after pX4_3 injection. Similar results were observed as with simultaneous injections (Supplementary Fig. 9e-f).

9. “eight PV patients versus eight healthy control skins” can you provide details of the clinical and immunological profile of these patients.

The clinical profile and immunological profile of the patient samples used have been added to Supplementary Table 4 (in the previous submission Supplementary Table 3).

10. “using PV-IgG and px4_3” can you characterize the immunophenotype of the PV IgG please

The ELISA values of the PV-IgG are detailed in the Methods section (Dsg1 1207 U/ml und Dsg3 3906 U/ml).

11. Figure 2d. why is HDAC3 activity going up over time in controls?

In the HDAC activity assay, a deacetylated fluorescent product is measured over time. As the product accumulates, the relative fluorescent units (RFU) increase over time. In PV-IgG treated cell extracts, HDACs are more active and more fluorescent product can accumulate than in IgG conditions. The panel to the right in Fig. 3e; and Supplementary Fig. 5b represents the slope of the curve multiplied by the protein amount as detailed in the Methods section.

Reviewer #2 (Remarks to the Author):

Franz et al performed a genome-wide Crispr/Cas9 screen in human keratinocytes to identify putative regulators of cell-cell adhesion using DSG3 expression as a readout. Moreover, the authors performed a Dsg3 promoter pull-down combined with MS approaches to identify Dsg3 promoter binding proteins. As outcome of both approaches, the authors identified KLF5 as a positive regulator of DSG3 expression and intercellular adhesion. Lowering KLF5 expression resulted in a diminished cell-cell adhesion of HaCaT cells, while enforced expression of KLF5 enhanced cell-cell adhesion. They further observed that KLF5 protein levels were reduced in skin biopsies of pemphigus vulgaris patient. Incubation of cell lines with PV-IgG autoantibodies resulted in the reduction of KLF5 expression and a parallel upregulation of the expression of HDAC3, which is a known KLF5 interactor, and in an increase in HDAC activity. The authors also observed acetylation of KLF5 in HaCaT cells. Furthermore, while pan HDAC inhibitors severely blocked HDAC activity in HaCaT cells, an HDAC3-specific inhibitor ameliorated the PV-IgG induced increased in HDAC activity, suggesting that HDAC3 is the primary HDAC family member activated in response to antibody treatment. Overexpression of HDAC3 reduced cell-cell adhesion of HaCaT cells, similar to sgRNA-mediated HDAC3 down-regulation, suggesting that HDAC3 expression levels have to be tightly controlled for the proper regulation of cell-cell adhesion. Finally, HDAC3 inhibitor treatment attenuates PV-IgG induced blister formation in mouse and human skin. Based on these data the authors conclude that KLF5 and HDAC3 are novel regulators of DSG3 gene expression and hence cell-cell adhesion. Overall, this is an interesting and nicely performed study. I have the following comments:

We thank the reviewer for the constructive review and suggestions and addressed his comments both experimentally and by discussion. Please find the point-to-point responses below:

(1) The authors showed that KLF5 is acetylated in HaCaT cells (SupFig.3d). Is the acetylation status regulated by HDAC3? Does overexpression of HDAC3 reduces KLF5 acetylation? And vice versa, does the application of an HDAC3 inhibitor increases KLF5 acetylation?

We evaluated KLF5 acetylation in more detail. However, we were not able to detect changes by PV-IgG using acetylation-specific KLF5 antibodies in Westerns (Fig. 4a). Moreover, KLF5 acetylation couldn't be reliably quantified by mass spectrometry which, however, might also result from technical limitations (not shown). Nevertheless, based on this we believe that changes of KLF5 acetylation do not play a major role in the responses to PV-IgG. However, PV-IgG incubation under conditions of increased HDAC3 protein levels suppressed KLF5 mRNA levels whereas HDAC3 depletion by sgRNA resulted in KLF5 mRNA upregulation (Fig. 4b), suggesting that HDAC3 is a regulator of KLF5 expression. A ChIP-qPCR experiment using

HDAC3 antibodies revealed a significant enrichment of HDAC3 at the KLF5 promoter (Fig. 4f). This was corroborated by analysis of published HDAC3 ChIP sequencing data from mouse epidermis (GSM4073696) (Supplementary Figure 6b). Moreover, treatment of HaCaT cells with PV-IgG antibodies increased HDAC3 on the KLF5 promoter of KLF5 (Fig. 4g). This suggests that, rather than acting by directly modulating KLF5 acetylation (as it was shown for HDAC1), HDAC3 functions as a repressor of KLF5 gene expression, specifically under PV-IgG conditions.

(2) Related to KLF5 acetylation status: lysine residue (K369) that are targeted by acetylation within KLF5 have been previously identified (e.g. PMID: 33731701). What happens if one overexpresses acetylation-deficient and acetylation-mimicking mutant forms (K to R and K to Q mutants, respectively)? Does overexpression of a mutant KLF5 still enhances DSG3 expression or even result in a higher expression compared to the effect from WT KLF5?

Although we detected that HDAC3 represses KLF5 transcription rather than acting as a deacetylase of KLF5, we additionally mutated two published acetylation sites of KLF5, K369 and K391, to alanine. While overexpression of wildtype KLF5 increased intercellular adhesion under basal conditions and in response to PV-IgG, the two acetylation-deficient mutants failed to do so (Fig. 3d, and Supplementary Fig. 6a). This suggests that, although HDAC3 does not alter KLF5 acetylation states, KLF5 acetylation promotes intercellular adhesion (Supplementary Fig. 6a).

3) Is the observed drop in KLF5 protein levels induced by PV-IgG treatment caused by reduced Klf5 gene expression or by posttranslational (or post transcriptional) changes? The authors should determine Klf5 gene expression levels in response to PV-IgG treatment. Moreover, to address KLF5 protein stability, the authors should perform cycloheximide experiments to study whether the half-life of KLF5 is changed in responses to PV-IgG treatment.

Thank you for this suggestion. The experiment was very helpful in understanding the mechanism how KLF5 is downregulated by PV-IgG (See comment 2). Quantitative real-time PCR of mRNA extracted from HaCaT cells treated with IgG or PV-IgG showed a significant downregulation of KLF5 mRNA using four different primer pairs (Fig. 4b). Since KLF5 is regulated at the mRNA level, we refrained from additionally performing experiments to test protein stability by blocking protein synthesis.

(4) Overexpression of HDAC3 resulted in the down-regulation of DSG3 (Fig. 3f), and down-regulation of HDAC3 also resulted in reduced DSG3 protein expression (Sup Fig. 3g). How can this be explained? HDAC3 also interacts with other factors as well as will class II HDACs. Could one explanation be that catalytic and non-catalytic functions (i.e. a scaffolding role) for HDAC3 depend on expression levels? Some enzymatic inactive forms of HDAC3 has been described (e.g. PMID: 24268577). What happens if one overexpresses a catalytically-dead HDAC3 version? Does this lead to a down-regulation of DSG3? And what happens if HDAC3 is down-regulated (instead of using an HDAC3 inhibitor) and cells are treated with PV-IgG?

Apparently, there is a slight misunderstanding. Overexpression of HDAC3 alone did not downregulate DSG3 but the levels rather stayed unchanged (previously Fig. 3f, now Supplementary Fig. 5a), while HDAC3 knockout in non-disease conditions results in reduced DSG3 levels. However, cell adhesion was impaired in both conditions. This suggests that different mechanisms leading to impaired adhesion are in place in the overexpression vs.

downregulation states. The notion, however, that under PV-IgG conditions (with a clear upregulation of HDAC3) HDAC3 inhibitors are effective and the cells are less prone for loss of cell-cell adhesion in sgHDAC3 cells, is indeed puzzling. One explanation could be that, under PV-IgG conditions, other HDAC3 mechanisms or effectors are triggered than under baseline conditions. While it would be very interesting to explore this further, this manuscript primarily focuses on the effects of HDAC3 and KLF5 modulation under PV-IgG conditions. We hope that the reviewer understands that for the sake of clarity and to keep the manuscript focused we refrained from putting more experimental effort in the role of HDAC3, specifically under baseline conditions. We adapted the results on pg. 9 and discussed these aspects on pg. 20-21.

“Application of HDAC3 inhibitors to enhance KLF5 stability, which we outlined in this study, may represent an alternative option to modulate DSG3 transcription. A weakness to this approach is the notion that both genetic HDAC3 deletion and overexpression resulted in impaired intercellular adhesion. This suggests that HDAC3 levels or activity need to be precisely calibrated to ensure intercellular adhesion. One reason for this behavior may be that, depending on the levels of HDAC3, different effector mechanisms are in place. For example, HDAC3 has functions requiring its catalytic domain while others rely on a scaffolding action (Emmett et al. PMID: 30390028; Sun et al. PMID: 24268577). Specifically in the context of PV-IgG, it is possible that primarily the catalytic function is enhanced (as indicated by the increased activity in HDAC activity assays), whereas in the long-term depletion context the non-catalytic functions are also affected. Alternatively, long-term HDAC3 depletion may trigger compensatory mechanisms which itself may affect intercellular adhesion. At least in the context of the short-term inhibition by RGFP966 or Entinostat, we did not observe a downregulation of DSG3 or loss of cell-cell adhesion under non-PV-IgG conditions, which might be in line with the notion that the inhibitors target the catalytic function of HDAC3 (Malvaez et al. PMID: 23297220). Although these limitations may reduce the therapeutic window in which a pharmacologic targeting of HDAC3 is beneficial, the strong upregulation in the PV setting harbors the potential to return HDAC3 and (through KLF5) also DSG3 to normal levels.”

(5) It is not clear, at least to this reviewer, why PV-IgG treatment results in the downregulation of KLF5 as shown in Figure 3c. Is there a signaling process induced upon binding of anti-DSG3 antibodies (PV-IgG)? If yes, does this mean that there is a feedback loop that result in the downregulation of KLF5 protein and hence a further decrease in DSG3? Since the authors observed that HDAC3 is upregulated upon PV-IgG binding (Figure 3e), do they propose a model in which PV-IgG treatment results in an upregulation of HDAC3 followed by a deacetylation of KLF5 and as a consequence reduced KLF5 stability? This is also related to my comment #1. The authors should consider to draw a model of how this might work.

It is established that signaling mechanisms are triggered by PV-IgG which contribute to disease development. Several kinases are activated upon PV-IgG addition and causally involved in loss of cell-cell adhesion, such as p38MAPK, ERK, Src and EGFR as discussed in the manuscript (pg. 22). We now added an experiment showing that inhibition of p38MAPK, a central pathway in PV, prevented the upregulation of HDAC3 in response to PV-IgG (Fig. 4h). Altogether, our results suggest a model in which under normal conditions KLF5 is expressed and acting as a transcriptional activator of DSG3. In the presence of PV-IgG, p38MAPK is activated and leads to high HDAC3 expression, which inhibits KLF5 expression at the

chromatin level. In the absence of KLF5, DSG3 levels are reduced, leading to reduced cell-cell adhesion and blister formation. A cartoon is now outlining this in Fig. 6g.

Reviewer #3 (Remarks to the Author):

Here, Franz et al identify KLF5 and HDAC3 as regulators of DSG3 protein level and cell adhesion, and positions them as possible therapeutic targets in PV. The topic is of interest to the field, the manuscript is extremely clear and well written, and the experiments well designed and clearly presented. As such this work represents an important contribution to the field. Several points below I believe would benefit from further discussion and exploration. These are not major and overall I find this work highly compelling.

We thank the reviewer for his positive response and additional suggestions to clarify and discuss specific aspects. Please find the point-to-point response below:

Line 164-168 “However, transduction of HaCaT cells with sgRNA targeting HDAC3 also resulted in a significant reduction of DSG3 protein levels (Fig 3g). This was accompanied by a significant decrease of cell-cell adhesion” . The authors note that this suggests precisely balanced activity, this notion deserves more discussion or exploration. I was comparing this with the pharmacological inhibition of HDAC3, which appears to show no significant change in adhesion or Dsg3 protein (Fig 4 a-b and Sup. Fig 4). This suggests something distinct between these two modes of suppressing HDAC3 which I would like the authors to discuss.

Thank you for the suggestion. We now discussed this topic in more detail on pg. 20-21:

“Application of HDAC3 inhibitors to enhance KLF5 stability, which we outlined in this study, may represent an alternative option to modulate DSG3 transcription. A weakness to this approach is the notion that both genetic HDAC3 deletion and overexpression resulted in impaired intercellular adhesion. This suggests that HDAC3 levels or activity need to be precisely calibrated to ensure intercellular adhesion. One reason for this behavior may be that, depending on the levels of HDAC3, different effector mechanisms are in place. For example, HDAC3 has functions requiring its catalytic domain while others rely on a scaffolding action (Emmett et al. PMID: 30390028; Sun et al. PMID: 24268577). Specifically in the context of PV-IgG, it is possible that primarily the catalytic function is enhanced (as indicated by the increased activity in HDAC activity assays), whereas in the long-term depletion context the non-catalytic functions are also affected. Alternatively, long-term HDAC3 depletion may trigger compensatory mechanisms which itself may affect intercellular adhesion. At least in the context of the short-term inhibition by RGFP966 or Entinostat, we did not observe a downregulation of DSG3 or loss of cell-cell adhesion under non-PV-IgG conditions, which might be in line with the notion that the inhibitors target the catalytic function of HDAC3 (Malvaez et al. PMID: 23297220). Although these limitations may reduce the therapeutic window in which a pharmacologic targeting of HDAC3 is beneficial, the strong upregulation in the PV setting harbors the potential to return HDAC3 and (through KLF5) also DSG3 to normal levels.”

In the mouse or human explant are there changes in Dsg3 protein level following PV which are rescued by inhibiting HDAC3? This result would be expected based on the cell culture work, however demonstration in the PV in vivo and ex vivo models would strengthen the mechanistic claims of the manuscript.

We agree with the reviewer and added this important experiment. In the ex vivo skin model, DSG3 levels decrease significantly after blister induction as indicated by DSG3 immunostaining. (Supplementary Figure 10a-b). Treatment with either RGFP966 or Entinostat ameliorated the reduction of DSG3 levels.

Another interesting question came to mind when reading this experiment. It is not clear if there would be different changes based on the epidermal layer – ie will inhibition of HDAC3 lead to expression of Dsg3 in the supra-basal layers of the epidermis? This could be tested, or added to the discussion of the manuscript.

Thank you for the suggestion. Dsg3 is expressed in the spinous layers and only absent in the granular layer. In the human skin model, we did not detect expression in the granular layer. However, we added this interesting aspect to the discussion on pg 21:

“While many silencing or knock-out approaches evaluated the effects of DSG3 loss on cellular functions, the long-term results of increasing DSG3 levels are largely unexplored. It is possible that this may interfere with layer-specific patterning of DSG3 expression in the epidermis which might lead to altered differentiation. For example, it was shown that forced expression of DSG3 in suprabasal epidermal layers resulted in differentiation defects (Merritt et al. PMID: 12138195). Moreover, it is known that desmosomal cadherins, at least to some extent, are regulated in a compensatory manner. For example, depletion of DSG3 leads to upregulation of DSG2 (Hartlieb et al. PMID: 24782306, Walter et al. PMID: 31178865). It is unclear whether such compensation effects are affected by long-term HDAC3 modulation and how this alters homeostasis in the setting of a differentiating epithelium. This, and the notion that HDAC3 inhibition affects the expression of a multitude of genes, will require careful dosing studies even though the skin is amenable to topical treatment.”

Finally, it is interesting to consider the roles of other desmosomal proteins. If Dsg3 is down regulated, are other cadherins up-regulated? The authors note in the discussion that KLF5 has been shown to regulate DSP, DSG1, and DSG2 in other tissues. Are these proteins also being regulated in keratinocytes upon inhibition of HDAC3?

We evaluated the levels of DSP and DSG1/DSG2 (using an antibody detecting both) in HaCaT. Inhibition of HDAC3 in HaCaT cells did not result in significant changes in DSP and DSG1/2 (Supplementary Fig. 6d). Moreover, KLF5 is not detected on the promoter regions of DSG2 in HaCaT cells according to a second screen we now added in comparison to DSG3 (Supplementary Figure 2a-b).

This is beyond the scope of the work, and I am not requesting an experiment, only discussion. Would treatment with an HDAC3 inhibitor could rescue an adhesion defect? Would there be an issue with altering DSG3 protein levels in healthy skin? Impacts on stratification? I appreciate this may be difficult to speculate on, but I found these questions exciting.

Thank you for the comments. We are not fully clear what the reviewer means with “Would treatment with an HDAC3 inhibitor could rescue an adhesion defect?” as we show that HDAC3 inhibition prevents loss of adhesion in the setting of PV-IgG. We however, now discuss the interesting question of a modulation of DSG3 levels in the setting of healthy skin. We now

included this aspect in the discussion on pg 21 (please see two comments above for the specific wording).

Minor

In Figure 3D there are two panels, the figure legend does not describe the left panel (RFU vs time).

The figure legend has been adapted and now reads: ". f) HDAC activity assay using HaCaT cell lysates. HaCaT cells were treated with Ctrl-IgG or PV-IgG including DMSO, 1 μ M trichostatin A (TSA) or 5 μ M RGFP966 for 24 hours. The left panel shows the increase in deacetylated peptide detected by its fluorophore over time of a representative experiment and the right panel shows the calculated HDAC3 activity (n=4).

REVIEWERS' COMMENTS

Reviewer #2 (Remarks to the Author):

The authors responded well to my comments.

Reviewer #3 (Remarks to the Author):

The authors have addressed all my concerns. The revised text and new experiments enhance the work, which is an important contribution to the field.

Reviewer #4 (Remarks to the Author):

This huge body of work is about finding new regulators of the DSG3 gene which could independently affect desmosomes and result in blistering and thus be targets for novel therapies for pemphigus.

A novel transcription factor called Kruppel-like-factor5 or KLF-5 was found to bind to the regulatory region of DSG3 to encourage adhesion.

In skin biopsies of PV patients, levels of KLF-5 were reduced, suggesting a role in the disease.

PV sera increased the levels of Histone deacetylase 3 (HDAC3) which reduced acetylation of KLF5 and induced blistering.

-PVIgG reduced levels of KLF5 mRNA (Fig 4b)

The implication was that KLF5 and HDAC3 could be new targets to treat pemphigus.

-

- Feedback No 1 - It would be useful to summarize in a table the various points of evidence, citing the figures, such as below:

Evidence that blocking HDAC3 reduces blistering

- BUT using sFRNA to HDAC3 reduced DSG protein levels and reduced cell to cell adhesion (supp 5d-e)

- CRISPR screen suggested HDAC3 as a positive reg of DSG3 in membrane (1B)

Both overexp and lower exp of HDAC3 gave reduced intercellular adhesion the correct BALANCE of this is needed for normal cohesion

- Titrating the inhibitor of HDAC3, RGFP966, to HaCaT cells incubated with PV-IgG , restored cell adhesion (Fig 5a) and with another inhibitor, Entinostat (fig 7a)

- Injection of these inhibitors after blistering induction improved the PV phenotype (supp 9 ef)

- Dsg3 levels were restored by inhibitors of HDAC3 (supp 10 ab)

- Ex vivo PV model of oral human mucosa – these inhibitors of HDAC3 improved the blistering induced by PX43

Evidence that enhancing HDAC3 increases blistering

- Human skin and mucosa in PV has increased HDAC3 on IF (figs 3g-h)

- HDAC3 depleted cells – when PV IgG added – no change in KLF5 (4d-e)

Evidence that increasing KLF5 reduces blistering

- KLF5 depleted HCT cells were used to test adhesion and could not be restored by HDAC3 (7e)

Evidence that blocking KLF5 increases blistering

KLF5 with mutated acetylation sites prevented loss of cell to cell adhesion after PV-IgG

PVIgG

Increases HDAC3 to increase blistering

- Shown in HaCaT cells with inc activity of HDAC3 -fig 3e

- with increased HDAC3 protein levels – fig 3f

- inhibitor of HDAC3 – RGFP966 – blocked the increased HDAC3 activity caused by PVIgG

- HDAC3 overexpression in HaCaT increased HDAC3 activity (supp 5b)

- reduced cell to cell adhesion (supplementary 5c) but Dsg3 levels were normal

Feedback No 2- Pemphigus patient material – please include information on list of patients – age, sex, race, site of skin or mucosa tissue

Feedback no 3- Diagrams that show the effects of the different factors on each other would be very helpful.

Reviewer #2 (Remarks to the Author):

The authors responded well to my comments.

Reviewer #3 (Remarks to the Author):

The authors have addressed all my concerns. The revised text and new experiments enhance the work, which is an important contribution to the field.

We thank reviewers #2 and #3 for assessing the manuscript as good for publication.

Reviewer #4 (Remarks to the Author):

This huge body of work is about finding new regulators of the DSG3 gene which could independently affect desmosomes and result in blistering and thus be targets for novel therapies for pemphigus. A novel transcription factor called Kruppel-like-factor5 or KLF-5 was found to bind to the regulatory region of DSG3 to encourage adhesion. In skin biopsies of PV patients, levels of KLF-5 were reduced, suggesting a role in the disease. PV sera increased the levels of Histone deacetylase 3 (HDAC3) which reduced acetylation of KLF5 and induced blistering. PVIgG reduced levels of KLF5 mRNA (Fig 4b). The implication was that KLF5 and HDAC3 could be new targets to treat pemphigus.

Feedback No 1 - It would be useful to summarize in a table the various points of evidence, citing the figures, such as below:

Evidence that blocking HDAC3 reduces blistering

- BUT using siRNA to HDAC3 reduced DSG protein levels and reduced cell to cell adhesion (supp 5d-e)

- CRISPR screen suggested HDAC3 as a positive reg of DSG3 in membrane (1B)

Both overexp and lower exp of HDAC3 gave reduced intercellular adhesion the correct BALANCE of this is needed for normal cohesion

- Titrating the inhibitor of HDAC3, RGFP966, to HaCaT cells incubated with PV-IgG, restored cell adhesion (Fig 5a) and with another inhibitor, Entinostat (fig 7a)

- Injection of these inhibitors after blistering induction improved the PV phenotype (supp 9 ef)

- Dsg3 levels were restored by inhibitors of HDAC3 (supp 10 ab)

- Ex vivo PV model of oral human mucosa – these inhibitors of HDAC3 improved the blistering induced by PX43

Evidence that enhancing HDAC3 increases blistering

- Human skin and mucosa in PV has increased HDAC3 on IF (figs 3g-h)

- HDAC3 depleted cells – when PV IgG added – no change in KLF5 (4d-e)

Evidence that increasing KLF5 reduces blistering

- KLF5 depleted HCT cells were used to test adhesion and could not be restored by HDAC3 (7e)

Evidence that blocking KLF5 increases blistering

KLF5 with mutated acetylation sites prevented loss of cell to cell adhesion after PV-IgG PVIgG

Increases HDAC3 to increase blistering

- Shown in HaCaT cells with inc activity of HDAC3 -fig 3e

- with increased HDAC3 protein levels – fig 3f

- inhibitor of HDAC3 – RGFP966 – blocked the increased HDAC3 activity caused by PVIgG

- HDAC3 overexpression in HaCaT increased HDAC3 activity (supp 5b)

- reduced cell to cell adhesion (supplementary 5c) but Dsg3 levels were normal

Thank you for the suggestion to provide an overview of the Figures. We have summarized the findings in the new supplementary table 6.

Feedback No 2- Pemphigus patient material – please include information on list of patients – age, sex, race, site of skin or mucosa tissue

Thank you for this suggestion. The manuscript already had included a table (Supp. Table 4) with information on age, sex and clinical data of the patients. In the course of this revision, we were asked to aggregate the information on age to ensure anonymity. Patients ranged in age from 34 to 97 years. We don't have information on the race of the patients. The remaining information is still present in Supp. Table 4.

Feedback no 3- Diagrams that show the effects of the different factors on each other would be very helpful.

Thank you for the suggestion. However, we are not sure what the reviewer specifically asks for. We already had included a graphical summary (Figure 6g). In our view, this, together with the new Supp. Table 6 (see Feedback #1) now should be sufficient to summarize the key findings of the manuscript.